

# Source Partitioning of $H_2O$ and $CO_2$ Fluxes Based on High Frequency Eddy Covariance Data: a Comparison between Study Sites

Anne Klosterhalfen[1], Alexander Graf[1], Nicolas Brüggemann[1], Clemens Drüe[2], Odilia Esser[1], María Pat González Dugo[3], Günther Heinemann[2], Cor M.J. Jacobs[4], Matthias Mauder[5], Arnold F. Moene[6], Patrizia Ney[1], Thomas Pütz[1], Corinna Rebmann[7], Mario Ramos Rodríguez[3], Todd M. Scanlon[8], Marius Schmidt[1], Rainer Steinbrecher[5], Christoph K. Thomas[9], Veronika Valler[1], Matthias J. Zeeman[5], Harry Vereecken[1]

[1]Agrosphere Institute, IBG-3, Forschungszentrum Jülich GmbH, 52425 Jülich, Germany
[2]Department of Environmental Meteorology, University of Trier, 54296 Trier, Germany
[3]IFAPA - Consejería de Agricultura, Pesca y Desarrollo Rural, Centro Alameda del Obispo, 14080 Córdoba, Spain
[4]Wageningen Environmental Research, Wageningen University and Research, 6708 PB Wageningen, the Netherlands
[5]Institute of Meteorology and Climate Research, IMK-IFU, Karlsruhe Institute of Technology (KIT), 82467 Garmisch-Partenkirchen, Germany
[6]Meteorology and Air Quality Group, Wageningen University and Research, 6708 PB Wageningen, the Netherlands
[7]Department Computational Hydrosystems, Helmholtz Centre for Environmental Research (UFZ), 04318 Leipzig, Germany
[8]Department of Environmental Sciences, University of Virginia, Charlottesville, VA 22904, United States
[9]Micrometeorology Group, University of Bayreuth, 95447 Bayreuth, Germany

*Correspondence to:* Anne Klosterhalfen (a.klosterhalfen@fz-juelich.de)

**Abstract.** For an assessment of the role of soil and vegetation in the climate system, a further understanding of the flux components of $H_2O$ and $CO_2$ (e.g., transpiration, soil respiration) and their interaction with physical conditions and physiological functioning of plants and ecosystems is necessary. To obtain magnitudes of these flux components, we applied the source partitioning approaches after Scanlon and Kustas (2010; SK10) and after Thomas et al. (2008; TH08) to high frequency eddy covariance measurements of twelve study sites including various ecosystems (croplands, grasslands, and forests) in a number of countries. Both partitioning methods are based on higher-order statistics of the $H_2O$ and $CO_2$ fluctuations, but proceed differently to estimate transpiration, evaporation, net primary production, and soil respiration. We compared and evaluated the partitioning results obtained with SK10 and TH08 including slight modifications of both approaches. Further, we analyzed the interrelations between turbulence characteristics, site characteristics (such as plant cover type, canopy height, canopy density and measurement height), and performance of the partitioning methods. We could identify characteristics of a data set as prerequisite for a sufficient performance of the partitioning methods.

SK10 had the tendency to overestimate and TH08 to underestimate soil flux components. For both methods, the partitioning of $CO_2$ fluxes was more irregular than of $H_2O$ fluxes. Results derived with SK10 showed relatively large dependencies on estimated water use efficiency (WUE) at leaf-level, which is needed as an input. Measurements of outgoing longwave radiation used for the estimation of foliage temperature and WUE could slightly increase the quality of the partitioning



results. A modification of the TH08 approach, by applying a cluster analysis for the conditional sampling of respiration/evaporation events, performed satisfactorily, but did not result in significant advantages compared to the other method versions (developed by Thomas et al., 2008). The performance of each partitioning approach was dependent on meteorological conditions, plant development, canopy height, canopy density, and measurement height. Foremost, the

performance of SK10 correlated negatively with the ratio between measurement and canopy height. The performance of TH08 was more dependent on canopy height and leaf area index. It was found, that all site characteristics which increase dissimilarities between scalars enhance partitioning performance for SK10 and TH08.

## 1 Introduction

The eddy covariance (EC) method is a micrometeorological technique commonly used to measure the energy, water vapor,
and carbon dioxide exchange between biosphere and atmosphere across a large range in time and space (Baldocchi et al., 2001; Reichstein et al., 2012). The measurements help to understand the temporal and spatial variations of these fluxes at the land-atmosphere interface. However, the EC method quantifies only net fluxes of water vapor, i.e., evapotranspiration (ET), and the net ecosystem exchange of $CO_2$ (NEE). Thus, for a better assessment of the role of soil and vegetation in the climate system, a further understanding of the flux components of $H_2O$ and $CO_2$ and their interaction with physical conditions and
physiological functioning of plants and ecosystems is necessary (Baldocchi et al., 2001). To obtain magnitudes of transpiration (T), evaporation (E), photosynthesis, and respiration by soil and vegetation, certain measurements with additional instrumentation independent of the EC technique can be conducted. Alternatively or additionally, so-called source partitioning approaches can be applied to the net fluxes obtained with the EC method. For instance, with the notion that during night no $CO_2$ is assimilated by plants (and hence observed NEE equals respiration), respiratory fluxes are often
estimated based on semi-empirical models describing the relationship between a physical driver (e.g., temperature) and respiration (Lloyd and Taylor, 1994; Reichstein et al., 2005, 2012). To estimate soil surface fluxes of both $H_2O$ and $CO_2$ directly from high frequency EC data without assumptions on such drivers, two new partitioning approaches were developed by Scanlon and coauthors (Scanlon and Sahu, 2008; Scanlon and Kustas, 2010), and Thomas et al. (2008). Both approaches rely on the assumption that the presence of multiple sources and sinks in and below the canopy will lead to decorrelation of
the high frequency scalar concentrations available in the framework of EC measurements above the canopy. This decorrelation contains information about the strength of these sinks and sources, which can be quantified by applying the flux-variance similarity theory or conditional sampling strategies. The scalar-scalar-correlations of $H_2O$ and $CO_2$ are however not only influenced by the sink-source distribution, but also by height (atmospheric surface layer, roughness sublayer), surface heterogeneity (Williams et al., 2007), canopy density, and coherent structures (Edburg et al., 2012; Huang
et al., 2013).

The source partitioning approach after Scanlon and Sahu (2008) and Scanlon and Kustas (2010) has been applied to data of a corn field (eastern USA; Scanlon and Kustas, 2012), compared to isotopic $H_2O$ flux partitioning (Good et al., 2014) and to





the Noah Land Surface Model (Wang et al., 2016) both for grasslands, and evaluated on a forest site on a decadal time scale (Sulman et al., 2016). Zeeman et al. (2013) further investigated the partitioning approach after Thomas et al. (2008) in association with coherent structures. To better assess these two approaches and their theoretical background, an intercomparison at a variety of study sites is necessary (Anderson et al., 2018).

The objective of this study is to compare and evaluate the source partitioning approaches after Scanlon and Kustas (2010) and after Thomas et al. (2008) by applying them to high frequency scalar measurements of various study sites in different ecosystems. Next to testing slight modifications of both partitioning methods, conditions and characteristics of study sites are identified under which the methods perform best. Based on findings of the above-mentioned authors and a large eddy simulation (LES) study (Klosterhalfen et al., in review), we hypothesize that the methods' performance is dependent on the

canopy height ($h_c$), which should represent the vertical separation of sinks and sources of $H_2O$ and $CO_2$ between canopy top and soil surface, on the canopy density (leaf area index LAI, LAI $h_c^{-1}$), and on the ratio between measurement height (z) and $h_c$. All these factors affect the degree of mixing of the scalars when they reach the EC sensors. With a high and sparse canopy and a low z $h_c^{-1}$, we assume a larger dissimilarity between scalar fluctuations and a more precise partitioning result of both source partitioning approaches. To summarize, goals of this study are:

- The comparison and evaluation of the partitioning results obtained with the approaches after Scanlon and Kustas (2010) and after Thomas et al. (2008) for various ecosystems, and testing slight modifications of the approaches

- An analysis of the two approaches with respect to their dependence on their underlying assumptions

- The description of the interrelations between turbulence characteristics, site characteristics (such as canopy type, $h_c$, z $h_c^{-1}$, LAI, and LAI $h_c^{-1}$), and performance of the partitioning methods

- The identification of characteristics of a data set (i.e. of study site and period properties), which lead to a satisfactory performance of the partitioning methods, if such characteristics exist.

## 2 Material and Methods

### 2.1 Source Partitioning after Scanlon and Kustas (2010) - SK10

To estimate the contributions of T, E, photosynthesis as net primary production (NPP), and soil respiration ($R_{soil,}$ autotrophic

and heterotrophic sources) to the measured net fluxes, Scanlon and Sahu (2008) and Scanlon and Kustas (2010) proposed a source partitioning method using high frequency time series from a typical EC station. This method (SK10 in the following) is based on the spatial separation and relative strength of sinks and sources of water vapor and $CO_2$ below the canopy (source of both water vapor and $CO_2$), in the canopy (source of water vapor and during daylight sink of $CO_2$), and the atmosphere. Assuming that air from those sinks and sources is not yet perfectly mixed before reaching EC sensors, partitioning is

estimated based on the separate application of the flux-variance similarity theory to the stomatal and non-stomatal components of the regarded scalars, as well as on additional assumptions on stomatal water use efficiency (WUE). The correlation between the two scalars ($\rho_{q'c'}$) usually deviates from -1 during daytime. This reduction of correlation and its




deviation from WUE at leaf-level is used to estimate the composition of the measured fluxes (Scanlon and Kustas, 2010; Scanlon and Sahu, 2008). For a detailed analytical description of SK10 see Scanlon and Albertson (2001), Scanlon and Sahu (2008), Scanlon and Kustas (2010, 2012), and Palatella et al. (2014). In the present study, SK10 was applied to high frequency EC data and the flux components were estimated as described by Klosterhalfen et al. (in review).

As mentioned before, the WUE at leaf-level has to be estimated for the application of SK10. WUE at leaf-level describes the relation between the amount of $CO_2$ uptake through stomata (photosynthesis) and the corresponding amount of $H_2O$ loss (T). One way to derive WUE (without additional measurements at leaf-level) is to relate the difference in mean $CO_2$ concentration between air outside and inside the leaf to the difference in mean water vapor concentration between air outside and inside the leaf including a factor that accounts for the difference in diffusion rate between $H_2O$ and $CO_2$ through the

stomatal aperture (Campbell and Norman, 1998; Scanlon and Sahu, 2008). The mean $H_2O$ and $CO_2$ concentrations just outside the leaf can be inferred from EC measurements by considering logarithmic mean concentration profiles implementing the Monin-Obukhov similarity theory (MOST; Scanlon and Kustas, 2010, 2012; Scanlon and Sahu, 2008). For the internal $CO_2$ concentration, a constant value of 270 or 130 ppm was presumed, typical for $C_3$ or $C_4$ plants, respectively (Campbell and Norman, 1998; Špunda et al., 2005; Williams et al., 1996; Xue et al., 2004). Values for the internal water

vapor concentration were estimated based on 100% relative humidity at foliage temperature ($T_f$). Measurements of $T_f$ were not available at the study sites, so for the source partitioning $T_f$ was set equal to measured air temperature ($WUE_{meanT}$; Scanlon and Sahu, 2008). Additionally, to investigate the sensitivity of WUE, $T_f$ was also derived by means of measured outgoing longwave radiation ($WUE_{OLR}$; with a surface emissivity of 0.98), or calculated similar to the external concentrations by considering a mean profile based on MOST ($WUE_{MOST}$). Thus, three different approaches to SK10 with

differing inputs for WUE were applied to all study sites.

## 2.2 Source Partitioning after Thomas et al. (2008) - TH08

Thomas et al. (2008) presented a new method (TH08 in the following) to estimate daytime sub-canopy respiration of forests directly from EC raw data by conditional sampling. At the same time, evaporation can be quantified by exchanging *c'* with *q'* in the equations given by Thomas et al. (2008, equations 1-11, pages 1212-1215). The method assumes that occasionally

air parcels moving upward (vertical wind fluctuations *w'* > 0) carry unaltered $H_2O$/$CO_2$ concentration combinations of the sub-canopy. Looking at the fluctuations *q'* and *c'*, both normalized with the corresponding standard deviation, respiration/evaporation signals should occur within the part of the joint probability distribution where *w'*, *q'* and *c'* are positive, i.e. in the first quadrant in the *q'-c'* plane. Additionally, Thomas et al. (2008) introduced a hyperbolic threshold criterion within quadrant 1, thus sampling all data points above this hyperbola. Thomas et al. (2008) found realistic

respiration estimates with a hyperbolic threshold of 0.25, which was also applied here. Subsequently, daytime evaporation and respiration can be estimated from the conditionally sampled *w'*, *q'*, and *c'* time series within quadrant 1 (Q1) or using the hyperbola threshold criterion (H). Furthermore, the turbulent $H_2O$ and $CO_2$ flux can be calculated by the covariance between *w'* and the corresponding scalar (CV) or represented by the relaxed eddy accumulation (REA) technique (Businger



and Oncley, 1990) using the coefficient $\beta$ as described in equation 4, page 1213 and statements on page 1215 in Thomas et al. (2008). Therefore, Thomas et al. (2008) applied four different approaches to quantify the respiration/evaporation events, combining the two conditional sampling criterions (Q1 or H) and the two calculation strategies (CV or REA technique).

For some averaging periods in our data, a potential respiration/evaporation 'cloud' was evident but did not occur within quadrant 1 (Fig. 1). As a modification of the conditional sampling strategy and a more tolerant detection of respiration/evaporation events, a distribution-based cluster analysis was conducted (fifth approach, GMM). With the Gaussian Mixture Model using the Expectation-Maximization Algorithm, two clusters or components, respectively, were defined for each averaging period: the respiration/evaporation 'cloud' and all further points associated with T and photosynthesis independent of the sign of $w'$. The GMM method is based on taking random samples from each component and fitting a certain number of Gaussian distributions to the samples, optimizing their parameters iteratively to model the data (Canty, 2010). Soil surface fluxes were calculated by CV from data in the respiration/evaporation 'cloud', where the deviations from the averages of all sampled cluster data points were used for $q'$ and $c'$ ($w'$ kept unchanged). Because the sampled respiration/evaporation 'cloud' by GMM would not always lie within quadrant 1 (often in quadrant 1 and 4, or in 1 and 2), and $q'$ and/or $c'$ of the defined 'cloud' could correlate negatively with $w'$, the corresponding surface flux would often be negative (Fig. 1). If this was the case for $H_2O$ and/or $CO_2$ soil fluxes, the corresponding flux was recalculated considering the deviations from the averages of all data points for $w'$, $q'$, and $c'$, and only including data points within quadrant 1 of the original $q'$-$c'$ plane and with $w' > 0$. This recalculated flux represented only a minimal fraction of the corresponding flux component. Also, as a result of this procedure the number of data points could differ between $H_2O$ and $CO_2$ for TH08 CV GMM depending on the used calculation step.

## 2.3 Study Sites and Data Processing

For the application and evaluation of the source partitioning methods, various study sites in a number of countries with differing cover types, canopy densities (regarding LAI), and measurement heights were chosen (Table 1). Almost all study sites are part of the FLUXNET network (Baldocchi et al., 2001). Detailed site and measurement descriptions can be found in the listed references. Next to coniferous and deciduous forests, grasslands, and croplands, some sites represent special canopy types: in SC_FR (for site abbreviations see Table 1) EC measurements have been conducted above an Mediterranean oak savanna (dehesa; Andreu et al., 2018); in Wüstebach an area of about 9 ha was deforested in 2013 and so measurements were obtained above the still present spruce forest (WU_FR) and the deforested area (WU_GL) (Graf et al., 2014; Wiekenkamp et al., 2016), where grass, shrubs, and young deciduous trees have been regrowing swiftly; and in LA_FR a coniferous forest has been regrowing gradually after a non-cleared windthrow in 2007 (Matiu et al., 2017). These three study sites represent the most heterogeneous landcover types in this study.

For each study site, measurements from days with a high-productive state of the vegetation and fair-weather conditions were selected to exclude factors interfering with the performance of the partitioning. Time periods with precipitation events were excluded. Furthermore, the quality assessment scheme after Mauder et al. (2013) was applied to each data set and source





partitioning was only conducted for time periods with the highest or intermediate quality flag levels assigned by this scheme. We only considered partitioning results of daytime data, because both methods require the presence of photosynthesis. Here, daytime was determined by calculating sunrise and sunset times by means of local time. Additionally, the TH08 method was only applied to time periods with a negative $\rho_{q'c'}$, and if less than 1% of the total data points in one half-hour time period

were sampled as the respiration/evaporation 'event', the partitioning result was disregarded.

The high frequency $H_2O$ and $CO_2$ time series of all study sites were pre-processed and prepared for the application of the source partitioning approaches as described by Klosterhalfen et al. (in review). For each study site, physically impossible values and spikes were excluded in the high frequency EC data of vertical wind, total $H_2O$ and $CO_2$ concentrations, the time delay was corrected, missing raw data within a half-hour period were gap-filled by linear interpolation, and a planar-fit

rotation was conducted, where the rotation matrix was calculated for only a maximum time period of two weeks. Further, the EC data was corrected for density fluctuations after Detto and Katul (2007). Then, the source partitioning approaches were applied to half-hourly time series of these pre-processed high frequency data, partitioning factors (E/ET or $R_{soil}$/NEE, respectively) were calculated, and applied to the post-processed half-hourly EC data.

## 2.4 Evaluation of Source Partitioning Results

The evaluation of the source partitioning performance was conducted in multiple ways for the various study sites depending on data-availability (Fig. 6). At some study sites, $R_{soil}$ was measured additionally with closed-chamber measurements independently of the EC measurements. In RO_GL and SE_CL, continuous measurements of multiple longterm-chambers were available for the considered time periods (half-hourly at SE_CL and hourly interpolated to half-hourly at RO_GL). In DI_CL, WU_FR, and WU_GL, $R_{soil}$ was measured with survey-chambers at several measurement points on one day during

the considered time periods, so spatial and temporal averages for the hours in question could be calculated. For all study sites (except for LA_FR), soil evaporation ($E_{soil}$) was estimated as a fraction of measured ET based on Beer's law depending on LAI ($E_{soil} = ET \exp(-0.6\ LAI)$; Campbell and Norman, 1998; Denmead et al., 1996). Thus, the root mean square error (RMSE) and the bias could be calculated between the partitioning results for E or $R_{soil}$ and the estimated $E_{soil}$ or chamber measurements, respectively. RMSE was sensitive to bias and outliers and the distribution of errors was skewed. The positive

outliers/errors (overestimations) were larger than negative errors (underestimations). A method overestimating the magnitude of a flux component may earn a larger RMSE than an underestimating one. Therefore, we also calculated a version of the RMSE based on log-transformed data ($RMSE_{ln}$; data transformed with $\ln(x+1)$) before computing differences between estimated and reference E or $R_{soil}$. Furthermore, one has to keep in mind that the measurements of $R_{soil}$ and LAI can also contain errors and that $E_{soil}$ is only a rough model approximation and can only give an order of magnitude to expect.

In addition, partitioned $CO_2$ fluxes were compared to results of the established partitioning approach after Reichstein et al. (2005), if available; even though this approach targets other flux components (total ecosystem respiration TER and gross primary production GPP). Here, the estimated NPP and $R_{soil}$ for every time step were classified as reasonable if their magnitudes were smaller than the calculated GPP or TER, respectively. Since all data sets were from the main growing




season and for weather conditions favorable to high respiration, we assumed that $R_{soil}$ should additionally be larger than 1 µmol m$^{-2}$ s$^{-1}$. In the following, NPP and $R_{soil}$ estimates meeting these criteria ("hits in range") will be counted as HiR GPP (magnitude of NPP smaller than magnitude of GPP) and HiR TER ($R_{soil}$ smaller than TER and larger than 1 µmol m$^{-2}$ s$^{-1}$). We calculated the relative fraction of HiR GPP and HiR TER in relation to the count of time steps with valid partitioning

solutions. Again, within this evaluation step two models including their different assumptions and uncertainties were examined and compared, and the results have to be handled with care. An evaluation of the estimated flux magnitudes was also possible for some study sites by means of former publications.

**2.5 Analysis of Source Partitioning Approaches**

To compare the strengths and limitations of SK10 and TH08 and to gain a better insight in their functionality and

dependencies on turbulence and site characteristics, a correlation analysis was conducted between HiR GPP or HiR TER and the variables $z$, $h_c$, $z\ h_c^{-1}$, LAI, or LAI $h_c^{-1}$. Here, we have chosen HiR GPP and HiR TER as the criteria of partitioning performance, because they could be calculated for all considered study sites, not like the error quantities (RMSE, bias, etc.) regarding $R_{soil}$. Different subsets of sites were considered to calculate the correlations: all study sites, only forest sites, or only cropland and grassland sites.

To obtain a better understanding of the strengths and limitations of TH08, we developed a conceptual model to generate simple, synthetic data sets of $w'$, $q'$, and $c'$ with different degrees of mixing between scalar sinks and sources from the soil, canopy, and boundary layer (Fig. 7, *upper panels*). We considered no mixing, complete mixing, and partial mixing between scalars originating from soil and canopy (with positive $w'$), all three sets excluding mixing with scalars originating from boundary layer (with negative $w'$). Averages of fluctuations were all specified as zero, and each scalar sink/source strength

was determined such that the net $H_2O$ flux equates to 1 mmol m$^{-2}$ s$^{-1}$ and the net $CO_2$ flux to -1 µmol m$^{-2}$ s$^{-1}$. To each generated data point of $q'$ and $c'$ a random number was added to simulate additional sources of variance not related to the degree of mixing. TH08 was applied to these synthetic data sets and could be validated with the true known partitioning factors, while SK10 was already thoroughly analyzed by means of the synthetic high frequency data derived by LES (Klosterhalfen et al., in review).

**3 Results and Discussion**

For each study site, the number of half-hourly time steps during daylight per considered time period is shown in Table A1 in the Appendix. Also, the relative fraction of daylight time steps of high-quality (HQ) which were used in the application of SK10 and TH08 are shown, where for SK10 only a good or intermediate quality flag (after Mauder et al., 2013) and no precipitation, and for TH08 additionally a negative $\rho_{q'c'}$ had to be given. Furthermore, the relative fraction of these HQ-time

steps, for which partitioning solutions were found, is shown for each method version. Thus, from the original data, only a part remained for the partitioning, and for only a part of this remaining data a partitioning result could be obtained.



### 3.1 Evaluation of Source Partitioning Results

In Fig. 2 the source partitioning results for $H_2O$ and $CO_2$ fluxes for LO_FR are shown in half-hourly time steps as an example. The partitioning results for all sites and all method versions are shown in the Supplementary material, including $E_{soil}$ estimations based on Beer's law, $R_{soil}$ chamber measurements, and/or partitioning results after Reichstein et al. (2005),

depending on data-availability. The diurnal dynamics of $H_2O$ and $CO_2$ fluxes, their components, and WUE and all method versions are shown in Fig. 3 for WA_FR. An overview of the partitioning results for all study sites is given in Fig. 4 using just two methods: SK10 with $WUE_{OLR}$ and TH08 with REA H. In Fig. 5 the total averages of the flux components over the available time periods are shown on the one hand comparing all method versions for a single site (MMP_FR) (*top panel*), and on the other hand comparing all sites for one method version (SK10 with $WUE_{OLR}$ and TH08 with REA H; *lower two*

*panels*). For the calculation of these diurnal dynamics and total averages large spikes in the estimated flux components (deviation from the mean by more than ten times of the standard deviation) were excluded. Figure 6 shows the error quantities, $RMSE_{ln}$ and bias relative to $R_{soil}$ chamber measurements, HiR GPP, HiR TER, and $E_{soil}$ estimation, for each site and method version. Timestamps in all following figures are in local time.

In general, SK10 and TH08 gave differing results for each study site and performed disparately between method versions. In

Fig. 2-5 it is apparent that TH08 mostly resulted in lower magnitudes of the flux components originating from the soil surface or sub-canopy, than SK10. The source partitioning results of LO_FR (Fig. 2, 5) were an exception to this rule. For this study site the partitioning fractions of SK10 and TH08 were very similar and thus suggest a very low uncertainty of the results. For the other study sites larger discrepancies were observed between SK10 and TH08.

For TH08, the calculation of the fluxes via REA yielded larger fluxes than via CV. Because averaging in the flux calculation

is performed differently for CV and REA (i.e. equations 1, page 1212 and equation 8, page 1214 in Thomas et al., 2008), and less data points are sampled with the hyperbolic threshold than using data from the entire Q1, the largest magnitudes were obtained by using REA with the hyperbolic threshold (REA H). In some time steps, no respiration/evaporation 'cloud' was apparent in the *q'-c'* plane, thus, the applied conditional sampling strategies could not be as effective as intended and an assessment of a correct sampling was not possible. Compared to the magnitude of GPP and TER estimated by the gap-filling

model after Reichstein et al. (2005), components estimated by TH08 almost always were within this prescribed range (magnitude of NPP smaller than magnitude of GPP, and $R_{soil}$ smaller than TER) because of their small resulting fluxes, whereby $R_{soil}$ was often below the assumed minimal threshold of 1 µmol m$^{-2}$ s$^{-1}$ and thus underestimated (Fig. 6, S1-S12 in Supplementary material). Regarding the error quantities in Fig. 6, TH08 REA H performed best. Partitioning results obtained by TH08 CV GMM were not systematically different from the other method versions but showed no extreme spikes in the

soil flux components.

The SK10 approach had the tendency to produce very high magnitudes of the soil flux components. Considering the diurnal dynamics and averages (Fig. 3-5), results of SK10 were satisfactory, but of course still relatively large. For most of the study sites, the magnitudes and variability in the half-hourly results of the soil flux components were decreased by using $WUE_{MOST}$



or $WUE_{OLR}$ instead of $WUE_{meanT}$. The differing WUE inputs had a larger effect on the $CO_2$ flux components than on $H_2O$. Considering the error quantities in Fig. 6, SK10 with $WUE_{OLR}$ very often gave the best results.

### 3.1.1 Evaluation by Means of Publications

The partitioned $CO_2$ fluxes generally showed a higher variability and more spikes than the partitioned $H_2O$ fluxes for all
sites. Jans et al. (2010) reported a mean $R_{soil}$ measurement of 3.16 μmol m$^{-2}$ s$^{-1}$ and a peak event of 23 μmol m$^{-2}$ s$^{-1}$ for DI_CL_MA in 2007. $R_{soil}$ estimates by SK10 were often as large as this peak, but the maximum observed by Jans et al. (2010) was triggered by precipitation, which does not apply for our considered time periods (Fig. S10 in Supplementary material). For LO_FR in 1997, Dolman et al. (2002) reported a peak respiration measurement of 12 μmol m$^{-2}$ s$^{-1}$, Falge et al. (2002) a seasonal maximum GPP of -24 μmol m$^{-2}$ s$^{-1}$ and seasonal maximum TER of 5.3 μmol m$^{-2}$ s$^{-1}$, and chamber
measurements from June 2003 had a magnitude of 17.3 μmol m$^{-2}$ s$^{-1}$. All these quantities support our partitioning results for LO_FR based on SK10, TH08, and the approach after Reichstein et al. (2005) (Fig. 2). For MMP_FR, Thomas et al. (2009) derived a T/ET ratio of 50% from sap flux measurements, which agrees well with the partitioning results by SK10 (Fig. S5 in Supplementary material). Results by TH08 and estimated $E_{soil}$ imply a larger fraction of T. For WU_GL, TH08 yielded results matching comparatively well to the modeled estimate $E_{soil}$ and the gap-filling approach after Reichstein et al. (2005)
(Fig. S8 in Supplementary material). As mentioned before, WU_GL is a very heterogeneous site with regrowing vegetation of grasses, shrubs, and trees on dry and wet areas. Thus, the measured signals could display fluxes coming from different sinks and sources distributed horizontally rather than vertically. The present variety of plant types increased the uncertainty in the estimation of WUE, where the usage of $WUE_{OLR}$ improved the partitioning by SK10 significantly, but the overestimation of $R_{soil}$ (compared to chamber measurements and TER) was not be avoided. For LA_FR we observed a
behavior similar to WU_GL. Here too, the forest is regrowing, but trees are already more abundant and larger. At SC_FR the impact of water stress on the application of source partitioning methods could be observed. For a very dry period in August 2016, both partitioning approaches were not applicable, because transpiration and photosynthesis almost ceased due to water stress and the correlations between $H_2O$ and $CO_2$ fluxes were almost always positive (not shown). For a period in April 2017, partitioning results could be obtained, where an increase in $R_{soil}$ estimated with SK10 and a decrease in estimated E
was evident during the respective time period (Fig. S6 in Supplementary material). Spring 2017 was considered as relatively dry in this region, and the last precipitation event was five days before the respective time period, so that it can be assumed that water stress increased steadily in April 2017. TH08 underestimated soil fluxes substantially, because no respiration/evaporation events were apparent, which could be caused by the sub-canopy in the oak savanna. For WA_FR, SK10-derived E and $R_{soil}$ were generally relatively large, only on 8 July 2016, the $CO_2$ flux components were smaller
(Fig. S3 in Supplementary material). On this day no significant differences in weather conditions or scalar statistics were apparent in contrast to the other days. In RO_GL the continuous $R_{soil}$ chamber measurements and TER estimated with the approach after Reichstein et al. (2005) did not agree well, where the latter decreased steadily over the seven days (this could





also be observed for FE_GL) and was mostly lower than measured $R_{soil}$. Compared to TER and measured $R_{soil}$, SK10 still overestimated and TH08 underestimated $R_{soil}$ fluxes.

### 3.1.2 Evaluation by Means of Error Quantities

A clear pattern in the performance of the source partitioning depending on method version or on study site characteristics could not be identified in the error quantities (Fig. 6). The following statements can be made:

1) The RMSE in $R_{soil}$ was usually larger for SK10 than for TH08 (not shown). Considering $RMSE_{ln}$ in $R_{soil}$, SK10 performed better at forest sites than TH08, and slightly worse at crop- and grasslands (Fig. 6a). The bias in $R_{soil}$ was always positive for SK10 (except for WU_FR) and often negative for TH08 (except for TH08 REA H; Fig. 6b). Therefore, SK10 has the tendency to overestimate and TH08 to underestimate $R_{soil}$ compared to respiration chamber measurements. The lowest RMSE, $RMSE_{ln}$, and bias were found for the SK10 method versions in WU_FR and for TH08 in WU_FR, WU_GL, and SE_CL_SB_09.

2) In comparison to the gap-filling model after Reichstein et al. (2005), HiR GPP were relatively frequent for TH08, with a minimum of 66.7% for SE_CL_SB_06, while HiR GPP for SK10 were usually less frequent. For HiR TER, both methods converged (Fig. 6c, d). While SK10 mostly overestimated TER, TH08 often estimated soil fluxes smaller than the minimal $R_{soil}$ threshold of 1 µmol m$^{-2}$ s$^{-1}$. TH08 REA H gave usually the best results for HiR TER and the worst for HiR GPP within the method versions of TH08. Also, the performance of SK10 improved for $CO_2$ in DI_CL_MA with increasing crop height and lower LAI (Fig. 4, 6).

3) The RMSE (not shown), $RMSE_{ln}$, and bias of E (compared to the modeled estimate $E_{soil}$ after Beer's law) were mostly similar or slightly larger for SK10 than for TH08 except for the low crop canopies, LO_FR, MMP_FR, and SC_FR (Fig. 6e, f). These sites also had a relatively low LAI. The error quantities were low in WU_FR and WU_GL for SK10 and TH08. The worst performance regarding E could be found in HH_FR for SK10, and in SC_FR, DI_CL_MA_06, and SE_CL_IC for TH08. The bias indicated that SK10 underestimated E for all canopies with a LAI lower than 2.3 (LO_FR, SC_FR, DI_CL_MA_06, SE_CL_SB_06, SE_CL_IC, the latter three have relatively short canopies). This could also be explained by the larger $E_{soil}$ estimates based on Beer's law due to the smaller LAIs, thus preventing an overestimation by SK10.

### 3.2 Analysis of Source Partitioning Approaches

### 3.2.1 Analysis by Means of Correlation Analysis

We studied the interrelations between partitioning performance (expressed in HiR GPP and HiR TER) and site characteristics such as canopy height $h_c$, LAI, canopy density (using LAI $h_c^{-1}$ as proxy), measurement height z, and the position of the measurements relative to the roughness sublayer (using z $h_c^{-1}$ as a proxy) by means of a correlation analysis (Tables 2, 3). Here, $h_c$ represents the vertical separation of sinks and sources of passive scalars between canopy top and soil



surface. As LAI can correlate with $h_c$ of a study site, LAI $h_c^{-1}$ was also considered to distinguish between their impacts on partitioning performance. The measurement height z was constant for each cropland, DI_CL_MA or SE_CL, for all considered time periods, thus the correlation coefficients with z have to be handled with care. All these geometric site characteristics represent some information on the characteristics of the turbulence and also affect the degree of mixing of the

scalars when they reach the EC sensor. Furthermore, we assume that with increasing LAI, LAI $h_c^{-1}$ and z $h_c^{-1}$, and with decreasing $h_c$ the dissimilarity between $q'$ and $c'$ decreases and EC measurements contain less information for the partitioning approaches (Edburg et al., 2012; Huang et al., 2013; Williams et al., 2007). Results of Klosterhalfen et al. (in review) suggest a decreasing performance of SK10 with increasing z $h_c^{-1}$.

Correlation coefficients between partitioning performance and site characteristics were calculated for all sites together, for

forests only, or for crop- and grasslands only, respectively (Tables 2, 3). For the SK10 method versions, the correlation coefficients showed similar relations between variables and partitioning results for both HiR GPP and HiR TER, because SK10 had the tendency to overestimate both NPP and $R_{soil}$. For the TH08 method versions, relations slightly differ between HiR GPP and HiR TER, because TH08 had the tendency to underestimate $R_{soil}$ fluxes ($< 1$ µmol m$^{-2}$ s$^{-1}$), thus HiR TER were smaller than HiR GPP. Only considering forest sites, the correlations were relatively high between variables and partitioning

performance.

The performance of all SK10 method versions correlated negatively with z $h_c^{-1}$ and positively with $h_c$ and z. The correlation coefficients regarding LAI, despite being also positive, were the smallest, where for the forest sites LAI was more important than for the remaining sites. LAI $h_c^{-1}$ correlated always negatively with performance of SK10 except for the forest sites, where the coefficients of LAI and LAI $h_c^{-1}$ were similar.

For the TH08 method versions LAI had larger, and $h_c$, z $h_c^{-1}$, and LAI $h_c^{-1}$ smaller effects on partitioning performance than for SK10 method versions. For HiR TER and only forests or only crop- and grasslands, $h_c$ was more important again in TH08 method versions (especially while neglecting the correlation with z). Correlation coefficients of LAI and LAI $h_c^{-1}$ were mostly positive with a few exceptions (e.g., regarding HiR TER for crop- and grasslands). For forest sites and TH08, only positive correlations were evident except for the relationship between HiR TER and z $h_c^{-1}$. Also, the impacts of $h_c$ and

LAI $h_c^{-1}$ were reversed between HiR GPP and HiR TER. Apparently, a dense forest canopy yielded too low sub-canopy fluxes derived by TH08, and a high canopy less reasonable canopy fluxes.

The variable LAI usually correlated positively for SK10 and TH08 method versions and all canopies, making it the sole variable which clearly contradicted our initial hypotheses. Also, the correlation between partitioning performance and LAI $h_c^{-1}$ at forest sites was contradictory. Next to canopy density, LAI could also be connected to larger sinks and sources of

canopy fluxes (T and photosynthesis) relative to soil surface fluxes due to larger biomass, and to the appearance and frequency of coherent structures. A dense canopy prevents frequent ejections of air parcels from the sub-canopy, but provokes higher scalar concentrations in such air parcels because of a longer accumulation under the canopy. Respiration/evaporation events could occur less frequent but be of higher magnitude. Also, small gaps in an otherwise dense canopy can play an important role regarding ejection events. Thus, how canopy density affects scalar-scalar-correlation



measured above the canopy (and associated with that the partitioning performance), cannot be easily assessed. In this study, canopy density and partitioning performance correlated negatively at crop- and grassland sites and positively at the forest sites. Assuming gaps in the canopy can be more expected in forests than in crop- or grasslands, these results support the above-mentioned aspects. For further assessments, an estimate about the (large-scale) heterogeneity and density of the

vegetation at study sites (gap fraction, canopy openness) has to be made and included in this analysis.

### 3.2.2 Analysis by Means of a Conceptual Model

SK10 was already thoroughly analyzed by means of the synthetic high frequency data derived by LES (Klosterhalfen et al., in review). In the present study, TH08 was applied to various synthetic $w'$-, $q'$-, and $c'$-data sets including soil, canopy, and boundary layer scalar sink/sources derived by a simple conceptual model as described above (Fig. 7, *top panel*). Defined by

the conditional sampling concept, we hypothesized that TH08 would work perfectly with no mixing of the scalars from the three different origins, would not obtain any partitioning factors in case of the complete mixing, and would underestimate the soil fluxes in case of partial mixing.

TH08 behaved as assumed except for TH08 REA H (see below; Fig. 7, *bottom panel*). For the partial mixing, a small difference in TH08-derived partitioning factors (especially for $H_2O$) was observed between the sampling in Q1 and with H,

because one data point was not sampled with the hyperbolic threshold, but laid within Q1. TH08 REA H did not yield any partitioning results in case of no or partial mixing. This is due to the different definitions of $\beta$ in the application of REA with the sampling in Q1 or with H (Thomas et al., 2008, equation 4, page 1213 and statement on page 1215). $\beta$ is an empirical constant and can be approximated by the ratio between the standard deviation of $w'$ ($\sigma_{w'}$) and the difference between the mean vertical velocities in updrafts and downdrafts ($\overline{w_+}$- $\overline{w_-}$). For the conditional sampling approach within Q1, $\beta$ is derived

including all data points (disregarding the sign of $q'$ or $c'$). For the approach including the hyperbolic threshold criterion, $\beta$ is derived from $w'$ data points which satisfy the hyperbolic threshold criterion for positive $q'$ and $c'$. In case of our conceptual model for the partial mixing, no data point with negative $w'$ satisfied this criterion, so without $\overline{w_-}$ $\beta$ and a partitioning factor could not be calculated. Figure 7 shows the partitioning factors for TH08 REA H while applying $\beta$ as calculated in TH08 REA Q1 (*non-filled markers*). TH08 CV GMM performed similar to the other method versions: sampled the correct

respiration/evaporation 'cloud' in case of no mixing and no 'cloud' in case of complete mixing. However, in case of the partial mixing all data points with $q' > 0$ were sampled by TH08 CV GMM, thus, considering also the fraction originating from the canopy. For the latter, the covariances applying the averages of $q$ or $c$ of the sampled cluster, and considering only data points with $w' > 0$, were negative for $H_2O$ and $CO_2$ (not shown). Thus, E and $R_{soil}$ were recalculated with the covariance taking the deviations of the average of $q$ or $c$ considering all data points, and including only data points with $w' > 0$, within

quadrant 1, and within the sampled cluster, thus, correcting the sampling by GMM, which resulted in a similar partitioning fraction as the other method versions.



## 4 Summary and Conclusions

The partitioning approaches after Scanlon and Kustas (2010; SK10) and after Thomas et al. (2008; TH08) gave differing results and performed disparately between method versions. TH08 mostly resulted in lower magnitudes of the flux components originating from the soil surface than SK10, and had the tendency to underestimate these flux components

compared to soil respiration flux measurements and estimates of $E_{soil}$ based on Beer's law. SK10 usually had the tendency to overestimate soil flux components and yielded larger error quantities (RMSE and bias), because the RMSE is depended on the bias and the error distribution was asymmetric. The positive errors (overestimations) were larger than negative errors (underestimations). Decreasing the weight of outliers by log-transforming $R_{soil}$ chamber observations and partitioning estimations revealed a lower $RMSE_{ln}$ for SK10 at forest sites than for TH08.

SK10 was used with a variety of estimates of WUE. Estimating input WUE using foliage temperature estimated from the observed outgoing longwave radiation often enhanced the partitioning performance. For TH08 various options where tested regarding the conditional sampling and flux calculation. Applying a Gaussian Mixture Model for the conditional sampling approach in TH08 did not improve partitioning performance significantly, because to obtain a positive and correct flux estimation was difficult from data points not within quadrant 1 in the $q'$-$c'$ plane. For TH08, conditional sampling including

a hyperbolic threshold and calculating flux components based on the relaxed eddy accumulation technique yielded the best partitioning results.

The partitioned $CO_2$ fluxes generally showed a higher variability and more spikes than the partitioned $H_2O$ fluxes for all sites and both methods. Also, mean diurnal cycles averaged over each site's regarded time period yielded satisfactory results for both approaches.

The dependencies of the partitioning performance on turbulence and site characteristics were analyzed based on a correlation analysis and the application of TH08 to synthetic, conceptual data sets of scalar fluctuations. Foremost, the performance of SK10 correlated negatively with the ratio between measurement height and canopy height. The performance of TH08 was more dependent on canopy height and leaf area index. Canopy density and partitioning performance of both methods correlated negatively at crop- and grassland sites and positively at the forest sites. All site characteristics which increase

dissimilarities between scalars enhance partitioning performance for SK10 and TH08.

For the forest site Loobos in The Netherlands, SK10 and TH08 obtained similar partitioning results and sufficient error quantities suggesting a low uncertainty. At this site with a relatively low LAI, high canopy, and low ratio between measurement and canopy height, conditions for both partitioning approaches seemed to be appropriate.

## Appendix A

30 In Table A1 the number of half-hourly time steps during daylight per considered time period is shown for each study site. Also, the relative fraction of daylight time steps of high-quality (HQ) which were used in the application of SK10 and TH08 are shown, where for SK10 only a good or intermediate quality flag (after Mauder et al., 2013) and no precipitation, and for





TH08 additionally a negative $\rho_{q'c'}$ had to be given. Furthermore, the relative fraction of these HQ-time steps, for which partitioning solutions were found, is shown for each method version. With TH08 by sampling in the first quadrant (Q1) a partitioning result could be obtained for almost every time step (minimum of 98.2%). With the hyperbolic threshold criteria and with GMM fewer solutions could be found, because quite often the number of sampled data points was less than 1% of

5 the total number in one half-hour time period. SK10 sometimes could not find a partitioning solution, when the measured and estimated $\rho_{q'c'}$ were not equal and removing large-scale processes by Wavelet-transform could not help either to solve the system of equations. The most solutions were found for MMP_FR (forest) and the least for RO_GL (grassland), suggesting a dependence on vegetation height. For crop sites DI_CL_MA and SE_CL_SB the number of solutions with SK10 increased with development stage of the maize or sugar beet, respectively, while the ratio between measurement

height and $h_c$ decreased. At the same sites the number of solutions for TH08 with hyperbolic threshold and GMM decreased (the conditional sampling in Q1 was not affected). Generally, for the grasslands and the lower crop canopies more solutions were obtained with TH08 than SK10. An exception was the low intercrop in Selhausen (SE_CL_IC).

*Supplement Link.*

*Competing interests.* The authors declare that they have no conflict of interest.

*Acknowledgements.* This research was supported by the German Federal Ministry of Education and Research BMBF, project IDAS-GHG [grant number 01LN1313A]. The measurement infrastructure providing observational data was supported by the German Research Foundation DFG through the Transregional Collaborative Research Centre 32 (TR 32) and Terrestrial Environmental Observatories (TERENO). The authors thank all technicians, engineers, and laboratory assistances in TR32, TERENO, and elsewhere for providing measurements of the test sites.

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




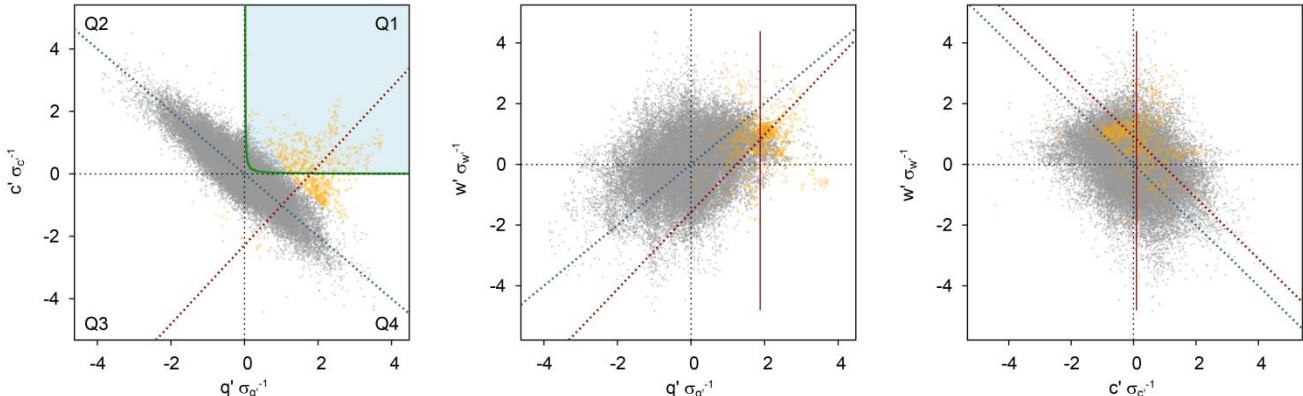

**Figure 1: Exemplary scatterplots of $w'$, $q'$, and $c'$ from WU_FR, 18 May 2015, 12:00-12:30 p.m. including results of the cluster analysis by Gaussian Mixture Model (orange data points) for the conditional sampling. Also shown are the hyperbolic threshold (H = 0.25, green line) after Thomas et al. (2008), the averages of $q$ and $c$ only considering data points of the respiration/evaporation 'cloud' (red lines), and reduced major axis regression lines after Webster (1997) for all data points (blue dashed lines) and only 'cloud' data points (red dashed lines).**

**In this example, calculating the covariance for $w$ and $c$ considering the $CO_2$ average of the 'cloud' yielded a negative soil flux (negative correlation). Thus, only 'cloud' data points within quadrant 1 in the original $q'$-$c'$ plane were considered for flux calculation using averages of all data points.**







**Figure 2: Source partitioning results of $H_2O$ (left) and $CO_2$ (right) fluxes in half-hourly time steps for the Loobos study site (forest) in The Netherlands and for every method version (see text for description). Grey areas show the measured water and $CO_2$ fluxes. Soil evaporation estimates derived based on Beer's law and $CO_2$ flux estimates by Reichstein et al. (2005; RE05) are also included**

5 **(LE: latent heat flux; E: evaporation; $E_{soil}$: estimated soil evaporation; GPP: gross primary production; NPP: net primary production; TER: total ecosystem respiration; $R_{soil}$: soil respiration; z: measurement height; $h_c$: canopy height; LAI: leaf area index).**





**Figure 3: Diurnal dynamics of source partitioning results of H₂O (left) and CO₂ (middle) fluxes and water use efficiency (WUE, right) for the Waldstein study site (forest) in Germany for 4-10 July, 2016 and for every method version (see text for description; LE: latent heat flux; E: evaporation; NPP: net primary production; R_soil: soil respiration; z: measurement height; h_c: canopy height; LAI: leaf area index). Error bars indicate the 95% confidence intervals of the mean values.**





**Figure 4: Diurnal dynamics of source partitioning results of $H_2O$ (upper panels) and $CO_2$ (lower panels) fluxes for all study sites and for the approach after Scanlon and Kustas (2010; SK10) with $WUE_{OLR}$ and after Thomas et al. (2008; TH08) with REA H (see text for description; LE: latent heat flux; E: evaporation; NPP: net primary production; $R_{soil}$: soil respiration). Error bars indicate the 95% confidence intervals of the mean values.**





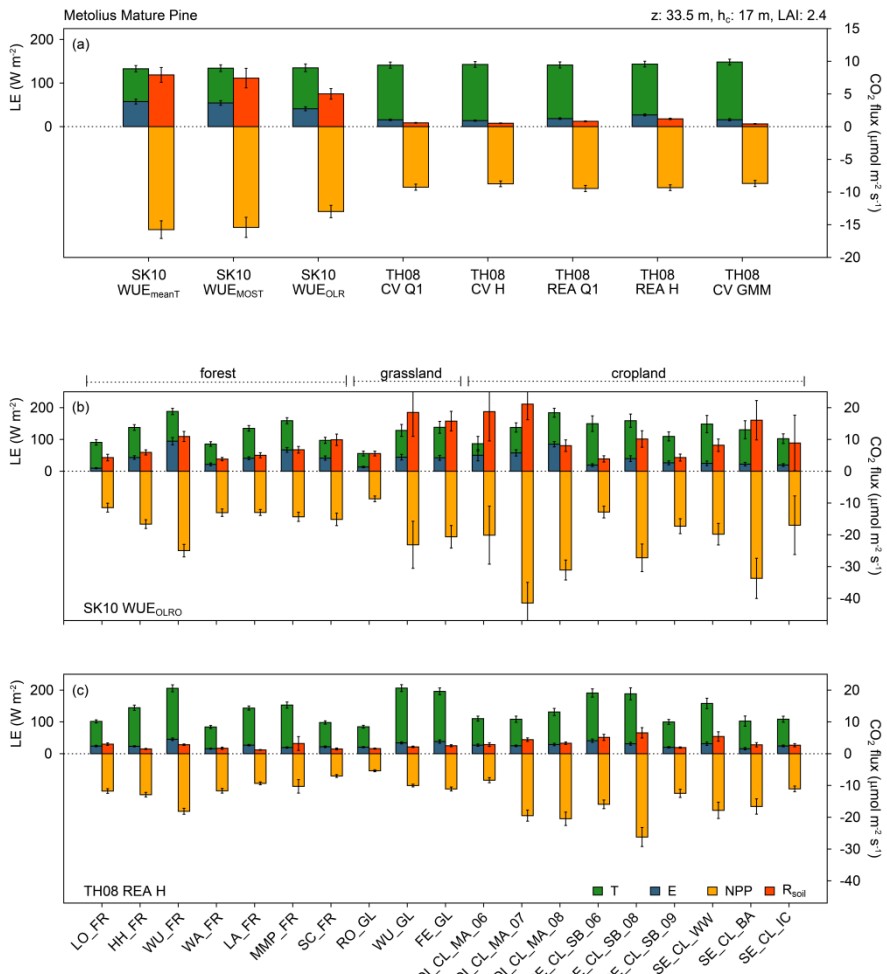

**Figure 5: Averages of source partitioning results of $H_2O$ and $CO_2$ fluxes, (a) for the Metolius Mature Pine study site (forest) in US and for every method version, (b) for all study sites and for the approach after Scanlon and Kustas (2010; SK10) with $WUE_{OLR}$, and (c) after Thomas et al. (2008; TH08) with REA H (see text for description; LE: latent heat flux; E: evaporation; NPP: net primary production; $R_{soil}$: soil respiration; z: measurement height; $h_c$: canopy height; LAI: leaf area index). Error bars indicate the 95% confidence intervals of the mean values. For each study site, net fluxes (evapotranspiration and net ecosystem exchange) differ between the two lower panels, because each method version found a different number of partitioning solutions, thus, the averages were taken from different subsets of the original data.**





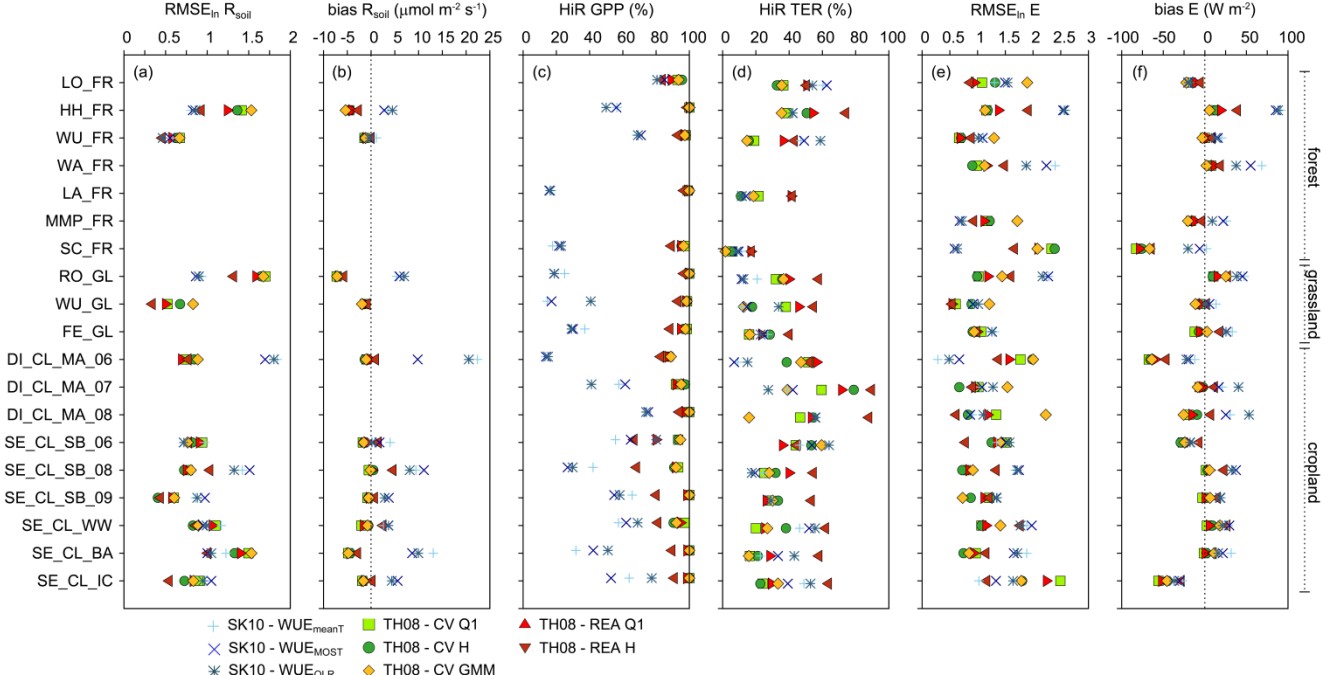

**Figure 6: Error quantities of source partitioning results for each study site and method version (see text for description). (a)-(b) Root mean square error in log-transformed data (RMSE$_{ln}$) and bias considering soil respiration (R$_{soil}$) chamber measurements, (c)-(d) relative fraction of time steps with partitioning results in range (HiR) of estimated gross primary production (GPP) and total ecosystem respiration (TER) by the approach after Reichstein et al. (2005), (e)-(f) RMSE$_{ln}$ and bias considering soil evaporation (E$_{soil}$) estimated based on Beer's law.**



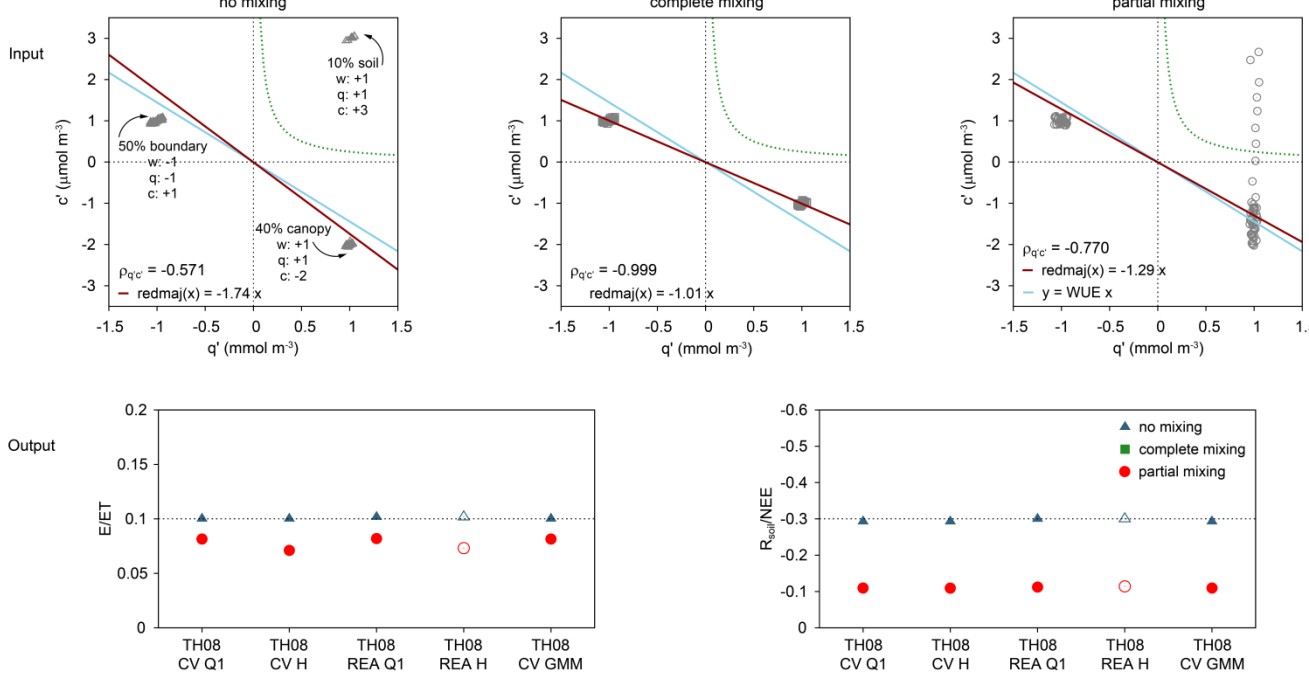

**Figure 7: Top: Setup of conceptual model for synthetic fluctuations (*q'* and *c'*) originating from soil, canopy, or boundary layer with differing degrees of mixing (no, complete, or partial mixing between soil and canopy sink/source) including water use efficiency (WUE = -1.444 µmol mmol$^{-1}$ = -3.53 mg g$^{-1}$; blue line), reduced major axis regression (red line) after Webster (1997), hyperbolic threshold criteria after Thomas et al. (2008; TH08) (H = 0.25; green dashed line) and correlation coefficient between *q'* and *c'* ($\rho_{q'c'}$). Bottom: True known partitioning ratios (dashed line) and source partitioning results of all TH08 method versions (see text for description) for each degree of mixing.**



**Table 1: Study sites and their characteristics (organized by first canopy type and second latitude; FR: forest; GL: grassland; CL: cropland).**

| abbrevi-ation | site | Latitude Longitude (m a.s.l.) | elevation | canopy type | period | LAI (m² m⁻²) | canopy height (m) | EC meas height (m) | mean annual Temp (°C) | mean annual P sum (mm a⁻¹) | prevailing wind direction | references |
|---|---|---|---|---|---|---|---|---|---|---|---|---|
| LO_FR | **Loobos** Gelderland, NL | 52.166581 5.743556 | 25 | **FR** (pine) | 08.-14. July 2003 | 1.9 | 18.6 | 24.0 | 10.0 | 966 | W-SW | Dolman et al., 2002 Elbers et al., 2011 |
| HH_FR | **Hohes Holz** ST, DE | 52.085306 11.222233 | 210 | **FR** (deciduous broadleaf) | 03.-09. July 2016 | 6.0 | 33.0 | 49.0 | 9.8 | 516 | SW | Wollschläger et al., 2017 |
| WU_FR | **Wüstebach (forest)** NRW, DE | 50.504907 6.331019 | 610 | **FR** (spruce) | 18.-24. May 2015 | 3.9 | 25.0 | 38.0 | 7.5 | 1220 | SSW | Ney et al., in review Graf et al., 2014 |
| WA_FR | **Waldstein** BY, DE | 50.14194 11.86694 | 775 | **FR** (spruce) | 04.-10. July 2016 | 5.5 | 25.0 | 36.0 | 5.8 | 885 | SSW | Babel et al., 2017 Foken et al., 2017 |
| LA_FR | **Lackenberg** BY, DE | 49.0996177 13.304667 | 1308 | **FR** (spruce/grass) | 24.-30. September 2017 | - | 3.0 | 9.0 | 3.7 | 1480 | SSW | Lindauer et al., 2014 Matiu et al., 2017 |
| MMP_FR | **Metolius Mature Pine** PNW, US | 44.4523 -121.5574 | 1253 | **FR** (pine) | 06.-12. June 2014 | 2.4 | 17.0 | 33.5 | 6.3 | 523 | SSW | Thomas et al., 2009 Vickers et al., 2012 |
| SC_FR | **Sta. Clotilde** ES | 38.210142 -4.287495 | 736 | **FR** (oak savanna) | 01.-07. April 2017 | 1.0 | 8.5 | 18.0 | 15.3 | 720 | SW | Andreu et al., 2018 |
| RO_GL | **Rollesbroich** NRW, DE | 50.621914 6.304126 | 515 | **GL** | 15.-21. July 2013 | 5.9 | 0.19 | 2.6 | 7.7 | 1033 | SSW | Borchard et al., 2015 Gebler et al., 2015 |
| WU_GL | **Wüstebach (clear cut)** NRW, DE | 50.503046 6.335946 | 610 | **GL** (deforested area) | 18.-24. May 2015 | < 2.5 | 0.25 | 2.5 | 7.5 | 1220 | SSW | Ney et al., in review Wiekenkamp et al., 2016 |
| FE_GL | **Fendt** BY, DE | 47.8329 11.0607 | 595 | **GL** | 11.-17. July 2015 | 3.5 | 0.25 | 3.5 | 8.4 | 1081 | SW | Zeeman et al., 2017 |
| DI_CL_MA_06 DI_CL_MA_07 DI_CL_MA_08 | **Dijkgraaf** Gelderland, NL | 51.992061 5.645944 | 9 | **CL** (maize) | 14.-16. June 2007 14.-16. July 2007 04.-06. August 2007 | 0.35 3.5 3.0 | 0.35 1.70 2.80 | 4.0 | 10.5 | 803 | S-SW | Jans et al., 2010 |
| SE_CL_WW SE_CL_BA SE_CL_IC SE_CL_SB_06 SE_CL_SB_08 SE_CL_SB_09 | **Selhausen** NRW, DE | 50.8658339 6.4473888 | 103 | **CL** (winter wheat) (barley) (intercrop) (sugar beet) | 03.-05. June 2015 27.-29. May 2016 23.-25. September 2016 20.-22. June 2017 02.-04. August 2017 04.-06. September 2017 | 6.1 5.1 1.0 2.3 5.2 4.3 | 0.79 0.95 0.22 0.37 0.46 0.50 | 2.4 | 9.9 | 698 | WSW | Eder et al., 2015 Ney and Graf, 2018 |





**Table 2: Correlation coefficients between partitioning performance of each method version regarding HiR GPP (see text for description) and study site characteristics ($h_c$: canopy height; LAI: leaf area index; z: measurement height) considering different sets of sites: all, only forest, or only crop- and grassland sites. Bold lettering indicates highest positive and highest negative correlation, and underlined (italic) lettering indicates highest (lowest) magnitude of correlation.**

| variable | SK10 $WUE_{meanT}$ | SK10 $WUE_{MOST}$ | SK10 $WUE_{OLR}$ | TH08 CV Q1 | TH08 CV H | TH08 REA Q1 | TH08 REA H | TH08 CV GMM |
|---|---|---|---|---|---|---|---|---|
| *all* | | | | | | | | |
| $h_c$ | **_0.52_** | **0.56** | **0.44** | 0.21 | 0.27 | 0.28 | 0.45 | 0.23 |
| LAI | *0.15* | *0.12* | *0.02* | **_0.43_** | 0.21 | **_0.43_** | 0.12 | **_0.26_** |
| z | 0.48 | 0.52 | 0.40 | 0.23 | **_0.27_** | 0.31 | **_0.48_** | 0.25 |
| z $h_c^{-1}$ | **-0.51** | **_-0.60_** | -0.45 | *-0.11* | -0.15 | -0.13 | -0.15 | *-0.10* |
| LAI $h_c^{-1}$ | -0.44 | -0.53 | **_-0.47_** | 0.19 | *0.05* | *0.11* | -0.11 | 0.12 |
| *forests* | | | | | | | | |
| $h_c$ | **0.64** | **0.63** | **0.56** | *0.20* | *0.21* | *0.21* | 0.27 | *0.11* |
| LAI | 0.35 | *0.32* | *0.26* | 0.61 | 0.74 | 0.68 | 0.70 | 0.65 |
| z | 0.62 | 0.60 | 0.55 | 0.37 | 0.31 | 0.36 | 0.41 | 0.27 |
| z $h_c^{-1}$ | **_-0.74_** | **_-0.75_** | **_-0.68_** | 0.27 | 0.25 | 0.28 | *0.20* | 0.37 |
| LAI $h_c^{-1}$ | *0.35* | 0.33 | 0.34 | **_0.77_** | **_0.78_** | **_0.81_** | **_0.83_** | **_0.79_** |
| *croplands, grasslands* | | | | | | | | |
| $h_c$ | **0.54** | **0.64** | **0.33** | 0.07 | **_0.23_** | 0.12 | 0.31 | 0.16 |
| LAI | 0.07 | *0.05* | *-0.10* | **0.40** | 0.10 | **_0.37_** | *-0.03* | 0.15 |
| z | *0.02* | 0.07 | -0.29 | **_-0.44_** | **-0.11** | **-0.17** | **_0.37_** | **-0.23** |
| z $h_c^{-1}$ | **_-0.58_** | **_-0.71_** | **_-0.51_** | *-0.01* | *-0.01* | *0.03* | 0.17 | *0.03* |
| LAI $h_c^{-1}$ | -0.37 | -0.49 | -0.46 | 0.37 | 0.21 | 0.32 | 0.16 | **_0.28_** |





**Table 3: Correlation coefficients between partitioning performance of each method version regarding HiR TER (see text for description) and study site characteristics ($h_c$: canopy height; LAI: leaf area index; z: measurement height) considering different sets of sites: all, only forest, or only crop- and grassland sites. Bold lettering indicates highest positive and highest negative correlation, and underlined (italic) lettering indicates highest (lowest) magnitude of correlation.**

| variable | SK10 $WUE_{meanT}$ | SK10 $WUE_{MOST}$ | SK10 $WUE_{OLR}$ | TH08 CV Q1 | TH08 CV H | TH08 REA Q1 | TH08 REA H | TH08 CV GMM |
|---|---|---|---|---|---|---|---|---|
| *all* | | | | | | | | |
| $h_c$ | **0.52** | **0.52** | **0.47** | -0.12 | -0.18 | **0.17** | *0.01* | -0.23 |
| LAI | *0.11* | *0.16* | *0.07* | -0.17 | **0.13** | *-0.02* | **0.33** | *-0.09* |
| z | 0.48 | 0.47 | 0.44 | **-0.17** | **-0.24** | 0.12 | -0.06 | **-0.27** |
| z $h_c^{-1}$ | **-0.47** | **-0.57** | -0.42 | **0.08** | *-0.01* | -0.14 | **-0.15** | **0.30** |
| LAI $h_c^{-1}$ | -0.42 | -0.50 | **-0.47** | *-0.08* | 0.03 | **-0.21** | -0.06 | 0.17 |
| *forests* | | | | | | | | |
| $h_c$ | **0.63** | **0.63** | 0.63 | **0.59** | **0.68** | 0.56 | 0.76 | **0.43** |
| LAI | 0.34 | 0.38 | *0.41* | 0.53 | 0.51 | **0.65** | **0.82** | 0.31 |
| z | 0.60 | 0.59 | **0.64** | 0.46 | 0.60 | 0.41 | 0.72 | 0.30 |
| z $h_c^{-1}$ | **-0.72** | **-0.73** | **-0.66** | -0.48 | -0.52 | -0.39 | *-0.47* | -0.35 |
| LAI $h_c^{-1}$ | *0.32* | *0.36* | 0.46 | *0.19* | *0.10* | *0.33* | 0.61 | *-0.11* |
| *croplands, grasslands* | | | | | | | | |
| $h_c$ | **0.54** | **0.59** | **0.34** | 0.42 | **0.61** | 0.50 | **0.85** | -0.25 |
| LAI | *0.01* | 0.06 | *-0.13* | **-0.49** | *-0.04* | **-0.33** | *0.03* | **-0.32** |
| z | 0.04 | *0.01* | -0.23 | **0.64** | 0.59 | **0.70** | 0.48 | *-0.03* |
| z $h_c^{-1}$ | **-0.48** | **-0.66** | -0.47 | *-0.16* | **-0.45** | *-0.20* | **-0.59** | 0.12 |
| LAI $h_c^{-1}$ | -0.34 | -0.47 | **-0.47** | -0.36 | -0.30 | -0.31 | -0.37 | -0.06 |




**Table A1:** Count of half-hourly time steps during daylight (CoD) per considered time period for each study site, corresponding relative fractions of CoD of high-quality (HQ) and relative fractions of these HQ-time steps with a found partitioning solution for each method version. Blue (red) lettering indicates the highest (lowest) fraction of solutions for each site. Bold (italic) lettering indicates the highest (lowest) fraction of solutions for each site. Superscript asterisk (minus) indicates the highest (lowest) fraction for each method version.

| method | site / time period | CoD | rel CoD used (HQ) | rel HQ with partitioning solution | site / time period | CoD | rel CoD used (HQ) | rel HQ with partitioning solution |
|---|---|---|---|---|---|---|---|---|
| SK10 WUE$_{meanT}$ | | | | 84.4 | | | | 26.2 |
| SK10 WUE$_{MOST}$ | | | 91.8 | 82.1 | | | 84.8 | 34.5 |
| SK10 WUE$_{OLR}$ | **LO_FR** | | | 65.6 | **DI_CL_MA_06** | | | *23.8* - |
| TH08 CV Q1, REA Q1 | 08.-14.07.2003 | 231 | | **99.4** | 14.-16.06.2007 | 99 | | **98.4** |
| TH08 CV H, REA H | | | 68.0 | 86.0 | | | 63.6 | 82.5 |
| TH08 CV GMM | | | | *59.2* | | | | 57.1 |
| SK10 WUE$_{meanT}$ | | | | 75.7 | | | | 90.4 |
| SK10 WUE$_{MOST}$ | | | 89.2 | 76.2 | | | 97.9 | 88.3 |
| SK10 WUE$_{OLR}$ | **HH_FR** | | | 74.8 | **DI_CL_MA_07** | | | 77.7 |
| TH08 CV Q1, REA Q1 | 03.-09.07.2016 | 231 | | **100.0** * | 14.-16.07.2007 | 96 | | **98.7** |
| TH08 CV H, REA H | | | 59.7 | 55.8 | | | 78.1 | *50.7* |
| TH08 CV GMM | | | | *51.4* | | | | 52.0 |
| SK10 WUE$_{meanT}$ | | | | 80.6 | | | | 95.3 * |
| SK10 WUE$_{MOST}$ | | | 78.0 | 78.8 | | | 94.5 | 94.2 |
| SK10 WUE$_{OLR}$ | **WU_FR** | | | 70.6 | **DI_CL_MA_08** | | | 89.5 |
| TH08 CV Q1, REA Q1 | 18.-24.05.2015 | 218 | | **100.0** * | 04.-06.08.2007 | 91 | | **100.0** * |
| TH08 CV H, REA H | | | 55.5 | 74.4 | | | 80.2 | *45.2* |
| TH08 CV GMM | | | | *51.2* | | | | 57.5 |
| SK10 WUE$_{meanT}$ | | | | 88.3 | | | | 57.3 |
| SK10 WUE$_{MOST}$ | | | 92.8 | 91.7 | | | 92.7 | 57.3 |
| SK10 WUE$_{OLR}$ | **WA_FR** | | | 89.3 | **SE_CL_SB_06** | | | 52.8 |
| TH08 CV Q1, REA Q1 | 04.-10.07.2016 | 222 | | **100.0** * | 20.-22.06.2017 | 96 | | **98.6** |
| TH08 CV H, REA H | | | 75.2 | 65.9 | | | 76.0 | 58.9 |
| TH08 CV GMM | | | | *50.3* | | | | *47.9* |
| SK10 WUE$_{meanT}$ | | | | *33.3* | | | | 72.9 |
| SK10 WUE$_{MOST}$ | | | 84.1 | 38.4 | | | 77.8 | 71.4 |
| SK10 WUE$_{OLR}$ | **LA_FR** | | | 56.5 | **SE_CL_SB_08** | | | 72.9 |
| TH08 CV Q1, REA Q1 | 24.-30.09.2017 | 164 | | **100.0** * | 02.-04.08.2017 | 90 | | **100.0** * |
| TH08 CV H, REA H | | | 54.9 | 93.3 * | | | 62.2 | *37.5* |
| TH08 CV GMM | | | | 58.9 | | | | 41.1 |
| SK10 WUE$_{meanT}$ | | | | 95.0 | | | | 80.6 |
| SK10 WUE$_{MOST}$ | | | 84.8 | 95.0 * | | | 92.3 | 81.9 |
| SK10 WUE$_{OLR}$ | **MMP_FR** | | | 93.3 * | **SE_CL_SB_09** | | | 81.9 |
| TH08 CV Q1, REA Q1 | 06.-12.06.2014 | 211 | | **100.0** * | 04.-06.09.2017 | 78 | | **98.3** |
| TH08 CV H, REA H | | | 73.0 | 70.8 | | | 76.9 | 25.0 - |
| TH08 CV GMM | | | | *60.4* | | | | *16.7* - |
| SK10 WUE$_{meanT}$ | | | | 73.9 | | | | 56.7 |
| SK10 WUE$_{MOST}$ | | | 87.4 | 75.2 | | | 93.8 | 52.2 |
| SK10 WUE$_{OLR}$ | **SC_FR** | | | 77.1 | **SE_CL_WW** | | | 46.7 |
| TH08 CV Q1, REA Q1 | 01.-07.04.2017 | 175 | | **99.3** | 03.-05.06.2015 | 96 | | **98.6** |
| TH08 CV H, REA H | | | 77.7 | 47.1 | | | 77.1 | *25.7* |
| TH08 CV GMM | | | | *40.4* | | | | 32.4 |



**Table A1 continued:**

| method | site / time period | CoD | rel CoD used (HQ) | rel HQ with part-itioning solution | site / time period | CoD | rel CoD used (HQ) | rel HQ with part-itioning solution |
|---|---|---|---|---|---|---|---|---|
| *SK10 WUE$_{meanT}$* | | | | *21.1 -* | **SE_CL_BA** | | | 50.6 |
| *SK10 WUE$_{MOST}$* | | | 91.7 | *32.7 -* | | | 82.3 | 51.9 |
| *SK10 WUE$_{OLR}$* | **RO_GR** | 217 | | 28.6 | 27.-29.05.2016 | 96 | | 58.2 |
| *TH08 CV Q1, REA Q1* | 15.-21.07.2013 | | | **100.0** * | | | | **98.5** |
| *TH08 CV H, REA H* | | | 73.3 | 53.5 | | | 67.7 | *26.2* |
| *TH08 CV GMM* | | | | 53.5 | | | | *27.7* |
| *SK10 WUE$_{meanT}$* | | | | *31.3* | **SE_CL_IC** | | | 64.6 |
| *SK10 WUE$_{MOST}$* | | | 82.1 | 38.0 | | | 91.5 | 70.8 |
| *SK10 WUE$_{OLR}$* | **WU_GR** | 218 | | 40.8 | 23.-25.09.2016 | 71 | | 73.8 |
| *TH08 CV Q1, REA Q1* | 18.-24.05.2015 | | | **100.0** * | | | | **98.2** - |
| *TH08 CV H, REA H* | | | 58.7 | 90.6 | | | 80.3 | 35.1 |
| *TH08 CV GMM* | | | | 88.3 * | | | | *28.1* |
| *SK10 WUE$_{meanT}$* | | | | *34.8* | | | | |
| *SK10 WUE$_{MOST}$* | | | 82.0 | 36.0 | | | | |
| *SK10 WUE$_{OLR}$* | **FE_GR** | 217 | | 39.9 | | | | |
| *TH08 CV Q1, REA Q1* | 11.-17.07.2015 | | | **100.0** * | | | | |
| *TH08 CV H, REA H* | | | 58.5 | 46.5 | | | | |
| *TH08 CV GMM* | | | | 65.4 | | | | |