# Peer review of "Source Partitioning of H2O and CO2 Fluxes Based on High Frequency Eddy Covariance Data: a Comparison between Study Sites"

_Biogeosciences, 2018_

## Short Comment (SC1) · 8 Nov 2018

I am just wondering this new method (in the following paper) can also be evaluated in this study.

Skaggs, T.H., Anderson, R.G., Alfieri, J.G., Scanlon, T.M., Kustas, W.P., 2018. Fluxpart: Open source software for partitioning carbon dioxide and water vapor fluxes. Agric For Meteorol, 253: 218-224. https://doi.org/10.1016/j.agrformet.2018.02.019

---

## Short Comment (SC2) · 14 Nov 2018

Thank you for your comment and suggestion! This other approach should indeed be referred to in any further version of our study. However, we understood that the main strength of the Skaggs et al. approach is in finding the optimal solution much faster than any earlier approach, rather than considerably changing results.

Our implementation of the source partitioning approach after Scanlon and Kustas (2010) resembles the procedure of Palatella et al. (2014) very closely. The only difference lays within terms of finding valid solutions with a minimal error in $\rho_{q'c'}$ and water use efficiency (WUE) at leaf-level. Because we converted Equation 15 in

[Figure]

Scanlon and Sahu (2008) solving for $\sigma^2_{c'_p}$ directly, we were left with only one unknown variable: $\rho_{c'_p c'_r}$. By insisting upon very low errors in $\rho_{q'c'}$ and WUE we found almost always the same solutions as the approach after Palatella et al. (2014), even though we passed on the implementation of the globally convergent Newton's method.

Skaggs et al. (2018) developed an algebraic solution simplifying the source partitioning procedure with a similar approach to ours. Using his analytical approach, we assume, would not change our results, but it might improve the rate of convergence to a solution. Preliminary tests done by Todd Scanlon comparing the approaches after Palatella et al. and Skaggs et al. support this: in addition to obtaining partitioning solutions for slightly more time steps, the new approach was more computationally efficient. Thus, in the face of an already extensive methodology, input variables and sites tested, we currently prefer the used approach and are confident that the impact on the results would be minimal.

Anne and Alex

---

## Referee Comment (RC1) · Anonymous Referee #1 · 27 Nov 2018

This manuscript presents a comparison of two turbulence-based flux partitioning methods across multiple sites representing a range of vegetation types (forest, grassland, and crop) and geographic zones. These emerging flux partitioning methods represent an effort to develop partitioning strategies that do not require assumptions about functional relationships, and this comparison between two methods across sites is a highly valuable contribution to the continuing development of new flux partitioning strategies. I have not seen a comprehensive comparison of two turbulence-based partitioning methods like this, and I think it represents an important step forward in understanding the performance of these methods. The comparison of multiple variations of each method and the analysis of specific site factors such as LAI and canopy height and how they

affect the methods are especially valuable contributions to development of these partitioning strategies. I thought the manuscript was clear, easy to follow, and well written overall. I only have a few comments for areas where the manuscript could be improved:

1. The manuscript refers several times to a manuscript by the same first author that is still in review in another journal. Until that manuscript is available to readers of this manuscript, I don't think it's useful to cite it. In particular, methodological details that have a bearing on this manuscript should be included in the supplemental material or main text, and not only cited to another manuscript that is not available at this time.

2. Tables 2 and 3 and A1 highlight the highest and lowest values of the metrics that they show. This makes it easy to ignore cases where there are multiple high values. It would be better to color code all the cells in the table based on their values, so readers could tell at a glance how the values looked. In addition, I think the correlations in Tables 2 and 3 should show whether they were statistically significant using bold text or asterisks.

3. The analysis used the ratio of LAI to canopy height as one of the predictors because "LAI can correlate with hc of a study site" (page 11, line 1). But LAI does not appear to be strongly correlated with hc for the sites in this study. Unless there is a strong relationship, this ratio seems difficult to interpret and I'm not sure I would include it in the analysis unless there is a clear interpretation.

Technical comments:

Page 7, line 5: Does "two models" refer to the two partitioning methods? They are not referred to as models elsewhere in the manuscript

Page 7, line 21: What distribution were the random numbers sampled from? Normal? If so, what were the mean and standard distribution?

Page 8, lines 7-8: This should include a brief explanation of why that site and those methods were chosen for the examples. Presumably because those methods had the

best performance?

Page 8, line 21: It should be "fewer data points"

Page 8, line 28: "TH08_REA_H performed best" needs more explanation. Based on what metric? Did it perform best for all sites and metrics, or a subset?

Page 9, line 3: The title of this section suggests that the following text will focus on comparing partitioning results to published analyses, but only a couple of the sites compare directly to publications. It might be more accurate to describe this as a detailed description of results for each site.

I think this paragraph should include a reference to Figure 5, since the bar plots are a helpful summary for many of the results described here.

I think this paragraph would be easier to follow if the supplementary figures were in the same order that they were referred to in the text.

Page 10, line 13-14: "both methods converged": It's not clear how they converged, or how that is shown in Fig. 6c and d.

Page 11, line 29: It's not clear how this was contradictory. Contradictory relative to what?

Figure 5: It is difficult to compare the two partitioning methods to each other across panels b and c. I suggest putting the two partitioning methods in the same panel so they can be directly compared, given the importance of these comparisons to the results. Perhaps panel b could show C fluxes and panel c could show LE, with bars for the two partitioning methods side-by-side in each panel.

Table 1: The abbreviations in the site column need to be defined (NL, ST, DE, PNL, . . .). Some of these are countries, some are regions, and some I didn't understand at all.

Page 29, line 3: The blue and red lettering is not in the table. As I said above, I think

color coding would be a good idea but it would be better to reflect the actual values rather than just where the highest/lowest value is.

---

## Referee Comment (RC2) · Anonymous Referee #2 · 28 Nov 2018

This study evaluates two approaches for partitioning eddy covariance fluxes into principle components (NPP and Soil respiration for carbon, and Transpiration and soil respiration for water). Both of the approaches (SK10 and TH08) rely on information contained in the raw, high frequency flux data, interpreted with assumptions about how the deviations in wind and gas concentrations should be correlated/coordinated for air parcels emerging from the canopy versus subcanopy. The developers of these approaches (Scanlon, Thomas) appear as co-authors on the paper, and the literature describing the approaches has been described elsewhere. Thus, while neither SK10 or TH08 is a perfect partitioning approach, I will focus my comments specifically on this effort to compare them (as opposed to comments about the underlying assumptions of

each).

I applaud the authors for this ambitious undertaking; it is not easy to handle raw data from so many flux sites. Methodologically (with one exception I'll address later), the work is sound. While it's may be a bit disappointing that the results weren't in better agreement, I think the paper contains information that will be of interest and useful to the flux community.

However, in its present form, I'm not certain that information is being successfully conveyed. Following are some comments on presentation, analysis, and methodology that may help to make the more accessible to others in the community who are seeking ways to better partition their tower-derived fluxes.

First, the paper is hard to read at times. This is due to many factors, including:

1. heavy reliance on acronyms,

2. very detailed explanation of methodology (i.e. the description of the 'GMM' approach on page 5),

3. Very nuanced description of some results that isn't organized around clear themes or patterns, (for example, the site-by-site analysis of performance in section 3.1.1),

4. some issues with grammar, and

5) a few very long paragraphs (i.e. page 9), and a few very short and choppy ones (page 13).

I urge the authors to carefully edit the writing with an eye towards: 1) moving information that is tangential to understanding the results to the SI (e.g. the GMM method description), 2) organizing results around clear patterns, and reducing words spent on detailed description of the site-by-site, or method-by-method results, and 3) carefully reviewing the text for language.

Second, the figures are also difficult to interpret, often because there are too many

panels. Some ideas for clearer presentation include:

Figure 2: Could the authors include fewer days of data, and perhaps consider omitting some of the different methods from the panel (for example, show TH08_REA_Q1 or TH08_REA_H, but not both). They seem quite similar.

Figure 3: Again, is it necessary to show each method's results?

Figure 4: Since you've already shown some of the diurnal dynamics, perhaps this figure could present daily-averages?

Figure 5: This figure is nice! It might be helpful (in a separate figure) to also show the estimated ratio of E:T, as this is often reported in the literature (see, for example, Good et al. 2015, Li et al. 2019).

Figure 6: Averaging across sites (or at least across plant functional types) would make it easier to understand the performance of the different partitioning methods.

Third, the authors focus most of their analysis on understanding differences in the magnitude of the partitioned fluxes (across a day, across sites). In my view, the magnitude of tower-derived fluxes will always be uncertainty, but as long as the sources of biases don't change too much in time, we can be more confident in using tower data to understand trends. How do these different partitioning methods agree in key functional relationships (for example, NPP versus PAR, Surface Conductance versus VPD)? Are the recovered trends as expected?

Fourth, I was confused by the HiP GPP and TER metric...it seems like the authors are filtering the data to consider only periods when the partitioned fluxes are similar in magnitude to those expected from conventional partitioning approaches (which are highly uncertain), and then using those filtered data to evaluated the partitioned fluxes? This seems like an approach that may obscure problems in one or the other partitioned fluxes...I would suggest a more straightforward comparison between the NPP and GPP (without the HiP) filtering.

Finally, are there any independent estimates of WUE (for example, from gas exchange data) in these sites, or similar biomes, that could provide a reality check on the tower-derived WUE estimates?

Work cited: Li et al. 2019. A simple and objective method to partition evapotranspiration into transpiration and evaporation at eddy-covariance sites. Agricultural and Forest Meteorology. https://www.sciencedirect.com/science/article/pii/S016819231830371X?via%3Dihub Good et al. 2015. Hydrologic connectivity constrains partitioning of global terrestrial water fluxes. Science. http://science.sciencemag.org/content/349/6244/175

---

## Author Comment (AC1) · 8 Jan 2019

*Thank you very much for your review of the abovementioned manuscript. We have carefully inspected all reviewer comments. Below, you will find our responses to the comments (italic) and we describe how we will try to implement the suggestions made by the reviewers. As primarily suggested by Reviewer #2, we will review our writing thoroughly for a better communication of our findings, if this manuscript is permitted to further revisions. We will also improve the figures as suggested by both Reviewers.*

*We hope that you will find the result satisfying.*

[Figure]

*Sincerely,*

*Anne Klosterhalfen, Alexander Graf, Nicolas Brüggemann, Clemens Drüe, Odilia Esser, María Pat González Dugo, Günther Heinemann, Cor M.J. Jacobs, Matthias Mauder, Arnold F. Moene, Patrizia Ney, Thomas Pütz, Corinna Rebmann, Mario Ramos Rodríguez, Todd M. Scanlon, Marius Schmidt, Rainer Steinbrecher, Christoph K. Thomas, Veronika Valler, Matthias J. Zeeman, and Harry Vereecken*

**Referee #1**

This manuscript presents a comparison of two turbulence-based flux partitioning methods across multiple sites representing a range of vegetation types (forest, grassland, and crop) and geographic zones. These emerging flux partitioning methods represent an effort to develop partitioning strategies that do not require assumptions about functional relationships, and this comparison between two methods across sites is a highly valuable contribution to the continuing development of new flux partitioning strategies. I have not seen a comprehensive comparison of two turbulence-based partitioning methods like this, and I think it represents an important step forward in understanding the performance of these methods. The comparison of multiple variations of each method and the analysis of specific site factors such as LAI and canopy height and how they affect the methods are especially valuable contributions to development of these partitioning strategies. I thought the manuscript was clear, easy to follow, and well written overall. I only have a few comments for areas where the manuscript could be improved:
*Thank you very much for this positive feedback.*

*1.1*

1. The manuscript refers several times to a manuscript by the same first author that is still in review in another journal. Until that manuscript is available to readers of this manuscript, I don't think it's useful to cite it. In particular, methodological details that have a bearing on this manuscript should be included in the supplemental material or main text, and not only cited to another manuscript that is not available at this time.

*The cited paper was accepted just recently and is now available online. We updated the reference in this manuscript.*

*1.2*

2. Tables 2 and 3 and A1 highlight the highest and lowest values of the metrics that they show. This makes it easy to ignore cases where there are multiple high values. It would be better to color code all the cells in the table based on their values, so readers could tell at a glance how the values looked. In addition, I think the correlations in Tables 2 and 3 should show whether they were statistically significant using bold text or asterisks.

*Our original versions of Tables 2, 3, and A1 were in color. But as far as we know (after contacting the Journal's Typesetting Department) tables in color are not possible.*

*In Tables 2 and 3, we also added asterisks for statistically significant correlations. Because the sample sizes were small and the data was often not normally distributed, the results have to be handled with care.*

*1.3*

3. The analysis used the ratio of LAI to canopy height as one of the predictors because "LAI can correlate with hc of a study site" (page 11, line 1). But LAI does not appear to be strongly correlated with hc for the sites in this study. Unless there is a strong relationship, this ratio seems difficult to interpret and I'm not sure I would include it in the analysis unless there is a clear interpretation.

*The ratio of LAI to canopy height ($h_c$) was used because it corresponds to plant area density. Considering only the study sites of one ecosystem type (forest, cropland, or grassland), correlations between LAI and $h_c$ can be found (see Fig. 1 below). For croplands, the correlation was weak, because for maize and sugar beet $h_c$ increased and LAI decreased with increasing maturity. Also, in this subset of sites in this particular study the maize crop in Dijkgraaf (DI_CL_MA_07 and DI_CL_MA_08) was a special case regarding its large (and expected) $h_c$. The correlation for grasslands was negative because of the very small sample size and different management strategies (dates of cutting) for each grassland, which influence both, LAI and $h_c$.*

*For clarification, we rephrased the following section: "For the chosen study sites, LAI correlated with $h_c$ when considering a specific ecosystem type (forest, cropland, or grassland). Thus, LAI $h_c^{-1}$ was also considered to distinguish between their impacts on partitioning performance."*

*We also think that this LAI-$h_c$-ratio may be useful for comparison to additional study sites. Thus, we would like to further include it in our analysis.*

*Fig. 1: Correlation between canopy height ($h_c$) and leaf area index (LAI) for each ecosystem type (FR: forest; CL: cropland; GL: grassland; R: Pearson product-moment correlation coefficient). Lines show reduced major axis regressions (after Webster 1997, European Journal of Soil Science 48:557).*

*1.4*
Technical comments:

*1.4.1*
Page 7, line 5: Does "two models" refer to the two partitioning methods? They are not referred to as models elsewhere in the manuscript
*Yes. For clarification, we rephrased the sentence to: "Again, within this evaluation step*

*two source partitioning approaches (SK10 or TH08 versus the approach after Reichstein et al., 2005) were examined and compared including their different assumptions and uncertainties,. . . ."*

**1.4.2**
Page 7, line 21: What distribution were the random numbers sampled from? Normal? If so, what were the mean and standard distribution?
*Yes. For clarification, we modified the sentence to: "To each generated data point of w', q' and c' a random number, sampled from a standard normal distribution and rescaled to a standard deviation of 5% of the magnitude of the variable, was added to simulate additional sources of variance not related to the degree of mixing."*

**1.4.3**
Page 8, lines 7-8: This should include a brief explanation of why that site and those methods were chosen for the examples. Presumably because those methods had the best performance?
*Done. We included following explanation: "In the following, figures are shown for some selected sites, which represent the overall results of all study sites the most, and/or for some selected method versions of SK10 and TH08, which usually presented the best partitioning performance."*

**1.4.4**
Page 8, line 21: It should be "fewer data points"
*Done.*

**1.4.5**
Page 8, line 28: "TH08 REA H performed best" needs more explanation. Based on

what metric? Did it perform best for all sites and metrics, or a subset?
*Done. We included following explanation: "Regarding the error quantities in Fig. 6,
TH08 REA H, among all TH08 method versions, yielded the best result for the largest
number of sites and error quantities."*

**1.4.6**
Page 9, line 3: The title of this section suggests that the following text will focus on
comparing partitioning results to published analyses, but only a couple of the sites
compare directly to publications. It might be more accurate to describe this as a
detailed description of results for each site.
*We changed the header to "Evaluation for Each Study Site" and we will try to include
further descriptions and valuable information of the partitioning results for missing sites.*

**1.4.7**
I think this paragraph should include a reference to Figure 5, since the bar plots are a
helpful summary for many of the results described here.
*Done.*

**1.4.8**
I think this paragraph would be easier to follow if the supplementary figures were in
the same order that they were referred to in the text.
*We organized the figures in the supplementary material as the study sites are listed in
Tab. 1 (organized by first canopy type and second latitude). Based on your comment,
we will try to reorganize the description and evaluation of the study sites in the text
after the same scheme.*

**1.4.9**

Page 10, line 13-14: "both methods converged": It's not clear how they converged, or how that is shown in Fig. 6c and d.

*For clarification, we rephrased the sentence as follows: "When using the gap-filling model after Reichstein et al. (2005) as a reference, high HiR GPP were relatively frequent for TH08, with a minimum of 66.7% for SE_CL_SB_06, while HiR GPP for SK10 were considerably lower (Fig. 6c). For HiR TER, such a clear difference in performance could not be observed (Fig. 6d)."*

*1.4.10*

Page 11, line 29: It's not clear how this was contradictory. Contradictory relative to what?

*We rephrased the sentence as follows: "Also, the correlation between partitioning performance and LAI $h_c^{-1}$ at forest sites contradicted our assumption that a higher plant density would have a negative effect."*

*1.4.11*

Figure 5: It is difficult to compare the two partitioning methods to each other across panels b and c. I suggest putting the two partitioning methods in the same panel so they can be directly compared, given the importance of these comparisons to the results. Perhaps panel b could show C fluxes and panel c could show LE, with bars for the two partitioning methods side-by-side in each panel.

*Done. We changed Fig. 5 as suggested in comment 1.4.11 and 2.2.4 by Reviewer #2.*

*1.4.12*

Table 1: The abbreviations in the site column need to be defined (NL, ST, DE, PNL,: : :). Some of these are countries, some are regions, and some I didn't understand at all.

*We adjusted the Tab. 1 mentioning only the countries. For a more pleasant reading, we will also change the acronyms of the study sites as suggested in comment 2.1 by*

*Reviewer #2.*

*1.4.13*
Page 29, line 3: The blue and red lettering is not in the table. As I said above, I think
color coding would be a good idea but it would be better to reflect the actual values
rather than just where the highest/lowest value is.
*Thank you for noticing this mistake. The reference to the blue and red lettering in the*
*table's caption was the description of the original colored table and was forgotten to be*
*removed while changing the table format (cf. comment 1.2). As far as we know, tables*
*in color cannot be included in this journal.*

*Thank you very much for your very constructive comments and your time!*

[Figure]

**Fig. 1.**

---

## Author Comment (AC2) · 8 Jan 2019

*Thank you very much for your review of the abovementioned manuscript. We have carefully inspected all reviewer comments. Below, you will find our responses to the comments (italic) and we describe how we will try to implement the suggestions made by the reviewers. As primarily suggested by Reviewer #2, we will review our writing thoroughly for a better communication of our findings, if this manuscript is permitted to further revisions. We will also improve the figures as suggested by both Reviewers.*

*We hope that you will find the result satisfying.*

*Sincerely,*

*Anne Klosterhalfen, Alexander Graf, Nicolas Brüggemann, Clemens Drüe, Odilia Esser, María Pat González Dugo, Günther Heinemann, Cor M.J. Jacobs, Matthias Mauder, Arnold F. Moene, Patrizia Ney, Thomas Pütz, Corinna Rebmann, Mario Ramos Rodríguez, Todd M. Scanlon, Marius Schmidt, Rainer Steinbrecher, Christoph K. Thomas, Veronika Valler, Matthias J. Zeeman, and Harry Vereecken*

**Referee #2**

This study evaluates two approaches for partitioning eddy covariance fluxes into principle components (NPP and Soil respiration for carbon, and Transpiration and soil respiration for water). Both of the approaches (SK10 and TH08) rely on information contained in the raw, high frequency flux data, interpreted with assumptions about how the deviations in wind and gas concentrations should be correlated/coordinated for air parcels emerging from the canopy versus subcanopy. The developers of these approaches (Scanlon, Thomas) appear as co-authors on the paper, and the literature describing the approaches has been described elsewhere. Thus, while neither SK10 or TH08 is a perfect partitioning approach, I will focus my comments specifically on this effort to compare them (as opposed to comments about the underlying assumptions of each).
I applaud the authors for this ambitious undertaking; it is not easy to handle raw data from so many flux sites. Methodologically (with one exception I'll address later), the work is sound. While it's may be a bit disappointing that the results weren't in better agreement, I think the paper contains information that will be of interest and useful to the flux community.

However, in its present form, I'm not certain that information is being successfully conveyed. Following are some comments on presentation, analysis, and methodology that may help to make the more accessible to others in the community who are seeking ways to better partition their tower-derived fluxes.
*Thank you very much for this constructive feedback.*

*2.1*
First, the paper is hard to read at times. This is due to many factors, including:
1. heavy reliance on acronyms,
2. very detailed explanation of methodology (i.e. the description of the 'GMM' approach on page 5),
3. Very nuanced description of some results that isn't organized around clear themes or patterns, (for example, the site-by-site analysis of performance in section 3.1.1),
4. some issues with grammar, and
5) a few very long paragraphs (i.e. page 9), and a few very short and choppy ones (page 13).
I urge the authors to carefully edit the writing with an eye towards: 1) moving information that is tangential to understanding the results to the SI (e.g. the GMM method description), 2) organizing results around clear patterns, and reducing words spent on detailed description of the site-by-site, or method-by-method results, and 3) carefully reviewing the text for language.
*We will review our writing thoroughly considering the above mentioned points.*

*2.2*
Second, the figures are also difficult to interpret, often because there are too many panels. Some ideas for clearer presentation include:

*2.2.1*

Figure 2: Could the authors include fewer days of data, and perhaps consider omitting some of the different methods from the panel (for example, show TH08_REA_Q1 or TH08_REA_H, but not both). They seem quite similar.

*Done. Fig. 2 shows now only 4 days of the considered time period and following methods: SK10 with $WUE_{meanT}$, $WUE_{MOST}$, and $WUE_{OLR}$, and TH08 CV Q1, REA H, and CV GMM (cf. comment 1.4.3 by Reviewer #1).*

*We added a figure for Loobos with results of all days and for every method version to the supplementary material.*

*2.2.2*
Figure 3: Again, is it necessary to show each method's results?
*Done. Fig. 3 shows now only following methods: SK10 with $WUE_{meanT}$, $WUE_{MOST}$, and $WUE_{OLR}$, and TH08 CV Q1, REA H, and CV GMM.*

*2.2.3*
Figure 4: Since you've already shown some of the diurnal dynamics, perhaps this figure could present daily-averages?
*With Fig. 4 we wanted to show at least once results of all study sites next to each other in the manuscript. Otherwise, we only show selected sites in the manuscript. We assume that daily averages would give a similar picture as Fig. 5.*

*2.2.4*
Figure 5: This figure is nice! It might be helpful (in a separate figure) to also show the estimated ratio of E:T, as this is often reported in the literature (see, for example, Good et al. 2015, Li et al. 2019).
*Thank you for this suggestion. Done. We changed Fig. 5 as suggested in comments 1.4.11 by Reviewer #1 and 2.2.4, also showing the partitioning factor E/ET. Also, we will include the suggested literature in our discussions comparing our partitioning*
*factors.*

*2.2.5*
Figure 6: Averaging across sites (or at least across plant functional types) would make it easier to understand the performance of the different partitioning methods.
*We agree that Fig. 6 is quite crowded, but averaging a performance metric/error quantity is not straightforward. It would probably require different strategies for the different error quantities and involve some arbitrary decisions. We see a high risk that the figure would be condensed at the cost of a much more difficult documentation of the methodology behind the figure. We would therefore prefer to keep it as it is.*

*2.3*
Third, the authors focus most of their analysis on understanding differences in the magnitude of the partitioned fluxes (across a day, across sites). In my view, the magnitude of tower-derived fluxes will always be uncertainty, but as long as the sources of biases don't change too much in time, we can be more confident in using tower data to understand trends. How do these different partitioning methods agree in key functional relationships (for example, NPP versus PAR, Surface Conductance versus VPD)? Are the recovered trends as expected?
*Thank you for this nice idea. We will have a closer look at such key functional relationships. Unfortunately, we cannot yet estimate, if such key functional relationships can be easily identified in our data because of too narrow ranges in the data or many additional and confounding factors (e.g., the relationship between NPP and PAR is also dependent on vegetation water status). As an example, Fig. 1 (below) shows the relationship between the averaged partitioning factor E/ET and LAI.*

*Fig. 1: Relationship between averaged partitioning factor E/ET (fraction of evaporation in evapotranspiration) and leaf area index LAI. Left diagram shows partitioning results*

[Figure]

of the method versions after Scanlon and Kustas (2010, SK10), and the right diagram of the method versions after Thomas et al. (2008, TH08). Green markers indicate forest sites, blue grassland sites, and yellow cropland sites.

**2.4**

Fourth, I was confused by the HiP GPP and TER metric...it seems like the authors are filtering the data to consider only periods when the partitioned fluxes are similar in magnitude to those expected from conventional partitioning approaches (which are highly uncertain), and then using those filtered data to evaluated the partitioned fluxes? This seems like an approach that may obscure problems in one or the other partitioned fluxes...I would suggest a more straightforward comparison between the NPP and GPP (without the HiP) filtering.

*We are sorry if the first manuscript version gave rise to a misunderstanding. The "Hit in Range" (HiR) criterion was solely used as one of three evaluation criteria (partitioning results in reference to Rsoil chamber measurements, HiR with respect to the approach after Reichstein et al. (2005), Esoil estimation according to Beer's law). It was NOT used to filter the data before any of the other analyses presented in the paper. We are aware that all of the abovementioned reference methods have their issues, which is why we used multiple of them and discuss them carefully.*

**2.5**

Finally, are there any independent estimates of WUE (for example, from gas exchange data) in these sites, or similar biomes, that could provide a reality check on the towerderived WUE estimates?

*We will conduct a more thorough literature search concerning estimates of WUE on leaf-level and extend our discussions. Unfortunately, no direct measurements of WUE were conducted at any study site.*

Work cited: Li et al. 2019. A simple and objective method to partition evapotranspiration into transpiration and evaporation at eddy-covariance sites. Agricultural and Forest Meteorology. https://www.sciencedirect.com/science/article/pii/S016819231830371X?via%3Dihub Good et al. 2015. Hydrologic connectivity constrains partitioning of global terrestrial water fluxes. Science. http://science.sciencemag.org/content/349/6244/175

*Thank you very much for your comments and your time!*

[Figure]

[Figure]

**Fig. 1.**

---

## Author Response (AR1)

**Biogeosciences Discussion - https://doi.org/10.5194/bg-2018-458**

Authors:     Anne Klosterhalfen, Alexander Graf, Nicolas Brüggemann, Clemens Drüe, Odilia Esser, María
             Pat González Dugo, Günther Heinemann, Cor M.J. Jacobs, Matthias Mauder, Arnold F. Moene,
             Patrizia Ney, Thomas Pütz, Corinna Rebmann, Mario Ramos Rodríguez, Todd M. Scanlon,
             Marius Schmidt, Rainer Steinbrecher, Christoph K. Thomas, Veronika Valler, Matthias J.
             Zeeman, Harry Vereecken,

Title:       Source Partitioning of $H_2O$ and $CO_2$ Fluxes Based on High Frequency Eddy Covariance Data: a
             Comparison between Study Sites.

**Response Letter**

*Dear Trevor Keenan, Dear Reviewers*

*Thank you very much for your review of the abovementioned manuscript. We have carefully inspected and replied to all reviewer comments and implemented the suggestions as documented in the following response letter. As primarily suggested by Reviewer #2, we revised our writing thoroughly for a better communication of our findings. We also improved most figures as suggested by both Reviewers (comments 1.4.11, 2.2).*

*The page and line numbers in the following reply refer to the revised manuscript version. Below, we included a version of the manuscript where all changes with respect to the previous version have been marked.*

*We hope that you will find the result satisfying.*

*Sincerely,*

*Anne Klosterhalfen, Alexander Graf, Nicolas Brüggemann, Clemens Drüe, Odilia Esser, María Pat González Dugo, Günther Heinemann, Cor M.J. Jacobs, Matthias Mauder, Arnold F. Moene, Patrizia Ney, Thomas Pütz, Corinna Rebmann, Mario Ramos Rodríguez, Todd M. Scanlon, Marius Schmidt, Rainer Steinbrecher, Christoph K. Thomas, Veronika Valler, Matthias J. Zeeman, and Harry Vereecken*

**Referee # 1**

This manuscript presents a comparison of two turbulence-based flux partitioning methods across multiple sites representing a range of vegetation types (forest, grassland, and crop) and geographic zones. These emerging flux partitioning methods represent an effort to develop partitioning strategies that do not require assumptions about functional relationships, and this comparison between two methods across sites is a highly valuable contribution to the continuing development of new flux partitioning strategies. I have not seen a comprehensive comparison of two turbulence-based partitioning methods like this, and I think it represents an important step forward in understanding the performance of these methods. The comparison of multiple variations of each method and the analysis of specific site factors such as LAI and canopy height and how they affect the methods are especially valuable contributions to development of these partitioning strategies. I thought the manuscript was clear, easy to follow, and well written overall. I only have a few comments for areas where the manuscript could be improved:

*Thank you very much for this positive feedback.*

**1.1**

1. The manuscript refers several times to a manuscript by the same first author that is still in review in another journal. Until that manuscript is available to readers of this manuscript, I don't think it's useful to cite it. In particular, methodological details that have a bearing on this manuscript should be included in the supplemental material or main text, and not only cited to another manuscript that is not available at this time.

*The cited paper was accepted just recently and is now available online. We updated the reference in this manuscript.*

**1.2**

2. Tables 2 and 3 and A1 highlight the highest and lowest values of the metrics that they show. This makes it easy to ignore cases where there are multiple high values. It would be better to color code all the cells in the table based on their values, so readers could tell at a glance how the values looked. In addition, I think the correlations in Tables 2 and 3 should show whether they were statistically significant using bold text or asterisks.

*Our original versions of Tables 2, 3, and A1 were in color. But as far as we know (after contacting the Journal's Typesetting Department) tables in color are not possible.*

*In Tables 2 and 3 (pages 29, 30), we also added asterisks for statistically significant correlations. Because the sample sizes were small and the data was often not normally distributed, the results have to be handled with care. For Forest_LA we received an estimate for LAI, which was missing before. Therefore, some results of the correlation analysis (Tab. 2, 3) differ from the previous manuscript version, but the main findings did not change.*

**1.3**

3. The analysis used the ratio of LAI to canopy height as one of the predictors because "LAI can correlate with hc of a study site" (page 11, line 1). But LAI does not appear to be strongly correlated with hc for the sites in this study. Unless there is a strong relationship, this ratio seems difficult to interpret and I'm not sure I would include it in the analysis unless there is a clear interpretation.

*The ratio of LAI to canopy height ($h_c$) was used because it corresponds to plant area density. Considering only the study sites of one ecosystem type (forest, cropland, or grassland), correlations between LAI and $h_c$ can be found (see Fig. R1 below). For forests, the correlation was low because of Forest_LA, where a dense spruce forest is regrowing after windthrow and the ratio between LAI and $h_c$ is similar to the ration in croplands. For croplands, the correlation was weak, because for maize and sugar beet $h_c$ increased and LAI decreased with increasing maturity. Also, in this subset of sites in this particular study the maize crop in Dijkgraaf*

*(Maize_DI_07 and Maize_DI_08) was a special case regarding its large (and expected) $h_c$. The correlation for grasslands was negative because of the very small sample size and different management strategies (dates of cutting) for each grassland, which influence both, LAI and $h_c$.*

*For clarification, we rephrased the following section: "For the chosen study sites, LAI correlated with $h_c$ when considering a specific ecosystem type (forest, cropland, or grassland). Thus, LAI $h_c^{-1}$ was also considered to distinguish between their impacts on partitioning performance" (page 11, line 24).*

*We also think that this LAI-$h_c$-ratio may be useful for comparison to additional study sites. Thus, we would like to further include it in our analysis.*

[Figure]

*Fig. R1: Correlation between canopy height ($h_c$) and leaf area index (LAI) for each ecosystem type (FR: forest; CL: cropland; GL: grassland; R: Pearson product-moment correlation coefficient). Lines show reduced major axis regressions (after Webster 1997, European Journal of Soil Science 48:557).*

1.4

Technical comments:

1.4.1

Page 7, line 5: Does "two models" refer to the two partitioning methods? They are not referred to as models elsewhere in the manuscript

*Yes. For clarification, we rephrased the sentence to: "Within this evaluation step two source partitioning approaches (approach after Reichstein et al., 2005 versus SK10 or TH08) were examined and compared including their different assumptions and uncertainties, ..." (page 7, line 10).*

**1.4.2**

Page 7, line 21: What distribution were the random numbers sampled from? Normal? If so, what were the mean and standard distribution?

*Yes. For clarification, we modified the sentence to: "To each generated data point of w', q' and c' a random number, sampled from a standard normal distribution and rescaled to a standard deviation of 5% of the magnitude of the variable, was added to simulate additional sources of variance not related to the degree of mixing" (page 7, line 28).*

**1.4.3**

Page 8, lines 7-8: This should include a brief explanation of why that site and those methods were chosen for the examples. Presumably because those methods had the best performance?

*Done. We included following explanation: "In the following, figures are shown for some selected sites, which were deemed most representative for all study sites, and/or for some selected method versions of SK10 and TH08, which usually exhibited the best partitioning performance" (page 8, line 10).*

**1.4.4**

Page 8, line 21: It should be "fewer data points"

*Done (page 10, line 33).*

**1.4.5**

Page 8, line 28: "TH08_REA_H performed best" needs more explanation. Based on what metric? Did it perform best for all sites and metrics, or a subset?

*Done. We included following explanation: "Regarding the error metrics in Fig. 6, TH08 REA H, among all TH08 method versions, yielded the best result for the largest number of sites and error metrics" (page 11, line 7).*

**1.4.6**

Page 9, line 3: The title of this section suggests that the following text will focus on comparing partitioning results to published analyses, but only a couple of the sites compare directly to publications. It might be more accurate to describe this as a detailed description of results for each site.

*Due to rewriting and restructuring this section '3.1 Evaluation of Source Partitioning Results' (as suggested in comment 2.1 by Reviewer #2), we evaluate the partitioning results on the one hand based on their flux magnitudes and in reference to former publications ('3.1.1 Flux Components Magnitudes'), and on the other hand based on error metrics in reference to chamber measurements, estimates of soil evaporation and the approach after Reichstein et al. (2005) ('3.1.2 Error Metrics') (pages 8-11).*

**1.4.7**

I think this paragraph should include a reference to Figure 5, since the bar plots are a helpful summary for many of the results described here.

*Done (e.g., page 8, line 31).*

**1.4.8**

I think this paragraph would be easier to follow if the supplementary figures were in the same order that they were referred to in the text.

*We organized the figures in the supplementary material as the study sites are listed in Tab. 1 (organized by first canopy type and second latitude). Based on your comment, we reorganized the description and evaluation of the study sites in the text after the same scheme (page 9, line 4).*

**1.4.9**

Page 10, line 13-14: "both methods converged": It's not clear how they converged, or how that is shown in Fig. 6c and d.

*For clarification, we rephrased the sentence as follows: "When using the gap-filling model after Reichstein et al. (2005) as a reference, high HiR GPP were relatively frequent for TH08, with a minimum of 66.7% for SugarBeet_SE_06, while HiR GPP for SK10 were considerably lower (Fig. 6c). For HiR TER, such a clear difference in performance could not be observed (Fig. 6d)" (page 10, line 17).*

**1.4.10**

Page 11, line 29: It's not clear how this was contradictory. Contradictory relative to what?

*We rephrased the sentence as follows: "Also, the correlation between partitioning performance by TH08 and LAI $h_c^{-1}$ at forest sites contradicted our assumption that a higher plant density would have a strong negative effect" (page 12, line 23).*

**1.4.11**

Figure 5: It is difficult to compare the two partitioning methods to each other across panels b and c. I suggest putting the two partitioning methods in the same panel so they can be directly compared, given the importance of these comparisons to the results. Perhaps panel b could show C fluxes and panel c could show LE, with bars for the two partitioning methods side-by-side in each panel.

*Done (page 25). We changed Fig. 5 as suggested in comment 1.4.11 and 2.2.4 by Reviewer #2.*

**1.4.12**

Table 1: The abbreviations in the site column need to be defined (NL, ST, DE, PNL,: : :). Some of these are countries, some are regions, and some I didn't understand at all.

*We adjusted Tab. 1 mentioning only the countries (page 28). For a more fluent reading, we changed the acronyms of the study sites as suggested in comment 2.1 by Reviewer #2.*

**1.4.13**

Page 29, line 3: The blue and red lettering is not in the table. As I said above, I think color coding would be a good idea but it would be better to reflect the actual values rather than just where the highest/lowest value is.

*Thank you for noticing this mistake (page 31). The reference to the blue and red lettering in the table's caption was the description of the original colored table and was forgotten to be removed while changing the table format (cf. comment 1.2). As far as we know, tables in color cannot be included in this journal.*

*Thank you very much for your very constructive comments and your time!*

**Referee # 2**

This study evaluates two approaches for partitioning eddy covariance fluxes into principle components (NPP and Soil respiration for carbon, and Transpiration and soil respiration for water). Both of the approaches (SK10 and TH08) rely on information contained in the raw, high frequency flux data, interpreted with assumptions about how the deviations in wind and gas concentrations should be correlated/coordinated for air parcels emerging from the canopy versus subcanopy. The developers of these approaches (Scanlon, Thomas) appear as co-authors on the paper, and the literature describing the approaches has been described elsewhere. Thus, while neither SK10 or TH08 is a perfect partitioning approach, I will focus my comments specifically on this effort to compare them (as opposed to comments about the underlying assumptions of each).

I applaud the authors for this ambitious undertaking; it is not easy to handle raw data from so many flux sites. Methodologically (with one exception I'll address later), the work is sound. While it's may be a bit disappointing that the results weren't in better agreement, I think the paper contains information that will be of interest and useful to the flux community.

However, in its present form, I'm not certain that information is being successfully conveyed. Following are some comments on presentation, analysis, and methodology that may help to make the more accessible to others in the community who are seeking ways to better partition their tower-derived fluxes.

*Thank you very much for this constructive feedback.*

**2.1**

First, the paper is hard to read at times. This is due to many factors, including:

**2.1.1**

1. heavy reliance on acronyms,

*For a more fluent reading, we changed the acronyms of the study sites (e.g., HH_FR to Forest_HH). Thus, Tab. 1 and A1, and Fig. 4, 5 and 6 had to be adjusted (pages 24-26, 28, 31-32). We have also refrained from using acronyms for the terms "foliage temperature" (page 4, lines 19-23).*

**2.1.2**

2. very detailed explanation of methodology (i.e. the description of the 'GMM' approach on page 5),

*We tried to shorten the indicated paragraph (page 5, lines 9-22). It describes a new conditional sampling technique and the subsequent flux calculation, so we try to explain our procedure completely and comprehensibly. Thus, we would not shorten the paragraph further.*

**2.1.3**

3. Very nuanced description of some results that isn't organized around clear themes or patterns, (for example, the site-by-site analysis of performance in section 3.1.1),

*We reorganized the description and evaluation of the partitioning results by first canopy type and second latitude (cf. comment 1.4.8 by Reviewer #1; page 9, line 4).*

**2.1.4**

4. some issues with grammar, and

*We reviewed our writing thoroughly and hope that all grammar mistakes etc. have been corrected.*

**2.1.5**

5) a few very long paragraphs (i.e. page 9), and a few very short and choppy ones (page 13).
*Based on your comment, we restructured most paragraphs.*

**2.1.6**

5   I urge the authors to carefully edit the writing with an eye towards: 1) moving information that is tangential to understanding the results to the SI (e.g. the GMM method description), 2) organizing results around clear patterns, and reducing words spent on detailed description of the site-by-site, or method-by-method results, and 3) carefully reviewing the text for language.
*We reviewed our writing thoroughly considering the above mentioned points and hope the result is satisfying.*

**2.2**

Second, the figures are also difficult to interpret, often because there are too many panels. Some ideas for clearer presentation include:

15 ### 2.2.1

Figure 2: Could the authors include fewer days of data, and perhaps consider omitting some of the different methods from the panel (for example, show TH08_REA_Q1 or TH08_REA_H, but not both). They seem quite similar.
*Done (page 22). Fig. 2 shows now only 4 days of the considered time period in Loobos and following methods:*
20   *SK10 with $WUE_{meanT}$, $WUE_{MOST}$, and $WUE_{OLR}$, and TH08 CV Q1, REA H, and CV GMM (cf. comment 1.4.3 by Reviewer #1).*
*We added a figure for Loobos with results of all days and for every method version to the supplementary material.*

25 ### 2.2.2

Figure 3: Again, is it necessary to show each method's results?
*Done (page 23). Fig. 3 shows now only following methods: SK10 with $WUE_{meanT}$, $WUE_{MOST}$, and $WUE_{OLR}$, and TH08 CV Q1, REA H, and CV GMM.*

30 ### 2.2.3

Figure 4: Since you've already shown some of the diurnal dynamics, perhaps this figure could present daily-averages?
*With Fig. 4 we wanted to show at least once results of all study sites next to each other in the manuscript (page 24). Otherwise, we only show selected sites in the manuscript. We assume that daily averages would give a*
35 *similar picture as Fig. 5.*

**2.2.4**

Figure 5: This figure is nice! It might be helpful (in a separate figure) to also show the estimated ratio of E:T, as this is often reported in the literature (see, for example, Good et al. 2015, Li et al. 2019).
40 *Thank you for this suggestion. Done (page 25). We changed Fig. 5 as suggested in comments 1.4.11 by Reviewer #1 and 2.2.4, also showing the partitioning factor E/ET. Also, we included the suggested literature in our discussions comparing our partitioning factors (page 8, line 31).*

**2.2.5**

45 Figure 6: Averaging across sites (or at least across plant functional types) would make it easier to understand the performance of the different partitioning methods.

*We agree that Fig. 6 is quite crowded (page 26), but averaging a performance metric / error quantity is not straightforward. It would probably require different strategies for the different error quantities and involve some arbitrary decisions. We see a high risk that the figure would be condensed at the cost of a much more difficult documentation of the methodology behind the figure. We would therefore prefer to keep it as it is.*

**2.3**

Third, the authors focus most of their analysis on understanding differences in the magnitude of the partitioned fluxes (across a day, across sites). In my view, the magnitude of tower-derived fluxes will always be uncertainty, but as long as the sources of biases don't change too much in time, we can be more confident in using tower data
10  to understand trends. How do these different partitioning methods agree in key functional relationships (for example, NPP versus PAR, Surface Conductance versus VPD)? Are the recovered trends as expected?

*Thank you for this nice idea. We had a closer look at such key functional relationships.*

*As an example, Fig. R2 (below) shows the relationship between the averaged partitioning factor E/ET and LAI for each study site and method version, where the E/ET derived by SK10 seems to be dependent on LAI. Fig. R3*
15  *(below) shows relationships between global radiation and hourly NPP, between air temperature and hourly $R_{soil}$, between vapor pressure deficit (VPD) and hourly T, and between VPD and estimated hourly, leaf-level WUE for the deforested area in Wüstebach (Grass_WU) for various method versions. The relationship between global radiation and estimated NPP showed a clear pattern for all method versions. For the other relationships (and for most of the study sites), no clear dependencies could be found in the hourly data because of too narrow data*
20  *ranges in the considered time periods (e.g., VPD only between 600 and 1200 Pa in Grass_WU) and many additional and confounding factors (e.g., the relationship between global radiation and NPP is also dependent on vegetation water status).*

*If desired, we can include these exemplary figures very gladly to the manuscript or Supplementary material (after some additional formatting) and discuss them in the manuscript.*

[Figure]

*Fig. R2: Relationship between averaged partitioning factor E/ET (fraction of evaporation in evapotranspiration) and leaf area index LAI. Left diagram shows partitioning results of the method versions after Scanlon and Kustas (2010, SK10), and the right diagram of the method versions after Thomas et al. (2008, TH08). Green markers*
30  *indicate forest sites, blue grassland sites, and yellow cropland sites.*

[Figure]

*Fig. R3: Relationships between global radiation and hourly net primary production (NPP), between air temperature and hourly soil respiration (R$_{soil}$), between vapor pressure deficit and hourly transpiration (T), and between vapor pressure deficit and estimated hourly, leaf-level water use efficiency (WUE) for the deforested area in Wüstebach (Grass_WU) for various method versions. Left column shows partitioning results of the method versions after Scanlon and Kustas (2010, SK10), and the right column of the method versions after Thomas et al. (2008, TH08).*

**2.4**

Fourth, I was confused by the HiP GPP and TER metric…it seems like the authors are filtering the data to consider only periods when the partitioned fluxes are similar in magnitude to those expected from conventional partitioning approaches (which are highly uncertain), and then using those filtered data to evaluated the partitioned fluxes? This seems like an approach that may obscure problems in one or the other partitioned fluxes…I would suggest a more straightforward comparison between the NPP and GPP (without the HiP) filtering.

*We are sorry if the first manuscript version gave rise to a misunderstanding. The "Hit in Range" (HiR) criterion was solely used as one of three evaluation criteria (partitioning results in reference to $R_{soil}$ chamber measurements, HiR with respect to the approach after Reichstein et al. (2005), $E_{soil}$ estimation according to Beer's law). It was NOT used to filter the data before any of the other analyses presented in the paper. We are aware that all of the abovementioned reference methods have their issues, which is why we used multiple of them and discuss them carefully.*

**2.5**

Finally, are there any independent estimates of WUE (for example, from gas exchange data) in these sites, or similar biomes, that could provide a reality check on the towerderived WUE estimates?

*We conducted a more thorough literature search concerning estimates of WUE at the leaf level and included references in our discussion (page 11, line 14). Unfortunately, no direct measurements of WUE were conducted at any study site.*

Work cited: Li et al. 2019. A simple and objective method to partition evapotranspiration into transpiration and evaporation at eddy-covariance sites. Agricultural and Forest Meteorology. https://www.sciencedirect.com/science/article/pii/S016819231830371X?via%3Dihub

Good et al. 2015. Hydrologic connectivity constrains partitioning of global terrestrial water fluxes. Science. http://science.sciencemag.org/content/349/6244/175

*Thank you very much for your comments and your time!*

[revised manuscript text omitted]
| Maize_DIA_06 | Dijkgraaf Netherlands, NL | 51.992106 5.645944 | 9 | CL (maize) | 14.-16. June 2007 | 0.35 | 0.35 | 4.0 | 10.5 | 803 | S-SW | Jans et al., 2010 |
| Maize_DIA_07 | | | | | 14.-16. July 2007 | 3.5 | 1.70 | | | | | |
| Maize_DIA_08 | | | | | 04.-06. August 2007 | 3.0 | 2.80 | | | | | |
| Wheat_SE_WW | Selhausen Germany, DE | 50.8658339 6.44743888 | 103 | CL (winter wheat) | 03.-05. June 2015 | 6.1 | 0.79 | 2.4 | 9.9 | 698 | WSW | Eder et al., 2015 |
| Barley_SE_BA | | | | (barley) | 27.-29. May 2016 | 5.1 | 0.95 | | | | | Ney and Graf, 2018 |

[revised manuscript text omitted]

(continued)

**Table A1 continued:**

| method | site time period | CoD | rel CoD used (HQ) | rel HQ with part-itioning solution | site time period | CoD | rel CoD used (HQ) | rel HQ with part-itioning solution |
|---|---|---|---|---|---|---|---|---|
| *SK10 WUE$_{meanT}$* | | | | *21.1* - | | | | 50.6 |
| *SK10 WUE$_{MOST}$* | | | 91.7 | *32.7* - | | | 82.3 | 51.9 |
| *SK10 WUE$_{OLR}$* | **Grass**_RO_ | 217 | | 28.6 | **Barley**_SE_ | 96 | | 58.2 |
| *TH08 CV Q1, REA Q1* | 15.-21.07.2013 | | | **100.0** + | 27.-29.05.2016 | | | **98.5** |
| *TH08 CV H, REA H* | | | 73.3 | 53.5 | | | 67.7 | *26.2* |
| *TH08 CV GMM* | | | | 53.5 | | | | 27.7 |
| *SK10 WUE$_{meanT}$* | | | | *31.3* | | | | 64.6 |
| *SK10 WUE$_{MOST}$* | | | 82.1 | 38.0 | | | 91.5 | 70.8 |
| *SK10 WUE$_{OLR}$* | **Grass**_WU_ | 218 | | 40.8 | **Intercrop**_SE_ | 71 | | 73.8 |
| *TH08 CV Q1, REA Q1* | 18.-24.05.2015 | | | **100.0** + | 23.-25.09.2016 | | | **98.2** - |
| *TH08 CV H, REA H* | | | 58.7 | 90.6 + | | | 80.3 | 35.1 |
| *TH08 CV GMM* | | | | 88.3 + | | | | *28.1* |
| *SK10 WUE$_{meanT}$* | | | | *34.8* | | | | |
| *SK10 WUE$_{MOST}$* | | | 82.0 | 36.0 | | | | |
| *SK10 WUE$_{OLR}$* | **Grass**_FE_ | 217 | | 39.9 | | | | |
| *TH08 CV Q1, REA Q1* | 11.-17.07.2015 | | | **100.0** + | | | | |
| *TH08 CV H, REA H* | | | 58.5 | 46.5 | | | | |
| *TH08 CV GMM* | | | | 65.4 | | | | |